# CRISPEDIT: Low-Curvature Projections for Scalable Non-Destructive LLM Editing

**Zarif Ikram** [1]  **Arad Firouzkouhi** [1]  **Stephen Tu** [1]  **Mahdi Soltanolkotabi** [1]  **Paria Rashidinejad** [1]

## Abstract

A central challenge in large language model (LLM) editing is capability preservation: methods that successfully change targeted behavior can quietly game the editing proxy and corrupt general capabilities, producing degenerate behaviors reminiscent of proxy/reward hacking. We present CRISPEDIT, a scalable and principled second-order editing algorithm that treats capability preservation as an explicit constraint, unifying and generalizing several existing editing approaches. CRISPEDIT formulates editing as constrained optimization and enforces the constraint by projecting edit updates onto the low-curvature subspace of the capability-loss landscape. At the crux of CRISPEDIT is expressing capability constraint via *Bregman divergence*, whose quadratic form yields the Gauss–Newton Hessian exactly and even when the base model is not trained to convergence. We make this second-order procedure efficient at the LLM scale using Kronecker-factored approximate curvature (K-FAC) and a novel *matrix-free projector* that exploits Kronecker structure to avoid constructing massive projection matrices. Across standard model-editing benchmarks and safety unlearning tasks, CRISPEDIT achieves high edit success while **keeping capability degradation below 1%** on average across datasets, significantly improving over prior editors.

## 1. Introduction

Large language models (LLMs) are rapidly becoming a shared backbone for knowledge work, spanning search and question answering (Gao et al., 2023b; Lewis et al., 2020),

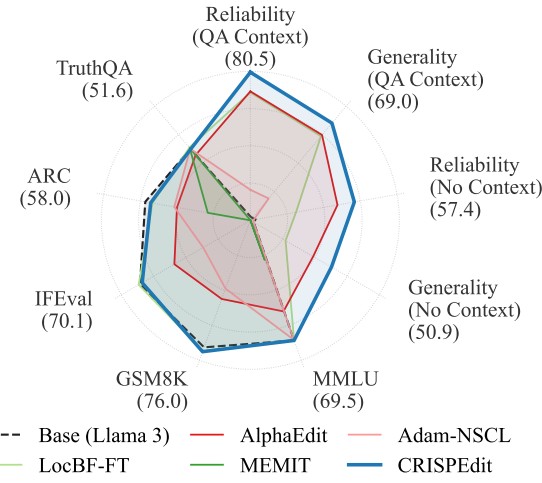

**Figure 1. Comparison overview of CRISPEDIT.** CRISPEDIT achieves strong edit reliability and generality, with and without QA context, while preserving broad base capabilities (MMLU, GSM8K, IFEval, ARC-C, TruthfulQA) on LLaMA-3-8B-Instruct.

science (Jumper et al., 2021), software development (Chen, 2021), decision support (Lopez-Lira & Tang, 2023), and education (Kasneci et al., 2023). Yet every day, facts shift, new discoveries land, products ship, hallucinations or unsafe behaviors are uncovered, quickly making the models stale. Retraining from scratch is the cleanest way to absorb this drift, but it is also the most expensive and slowest. Model editing (Sinitsin et al., 2020; De Cao et al., 2021; Mitchell et al., 2022b; Wang et al., 2024c) offers a practical alternative: apply targeted updates to correct a fact, insert new knowledge, remove unsafe behavior, personalize the style *while leaving everything else intact*.

In many cases, edits may appear to succeed while quietly degrading broader capabilities reminiscent of reward/proxy hacking (Gao et al., 2023a). This degradation can manifest as brittle reasoning, weaker instruction-following, or even broken fluency. This failure mode is especially pronounced for safety-oriented edits, such as unlearning private information or copyright-sensitive content, where apparent success may reflect broad refusal or capability collapse rather than precise removal of the targeted behavior. In response, prior work has introduced heuristic guardrails: restrict updates to a small set of parameters (Hu et al., 2022; Yu et al., 2024), localize "where the knowledge lives," (Meng et al.,

---

[1]University of Southern California. Correspondence to: Zarif Ikram <zikram@usc.edu>, Paria Rashidinejad <paria.rashidinejad@usc.edu>.

*Proceedings of the 43rd International Conference on Machine Learning*, Seoul, South Korea. PMLR 306, 2026. Copyright 2026 by the author(s).

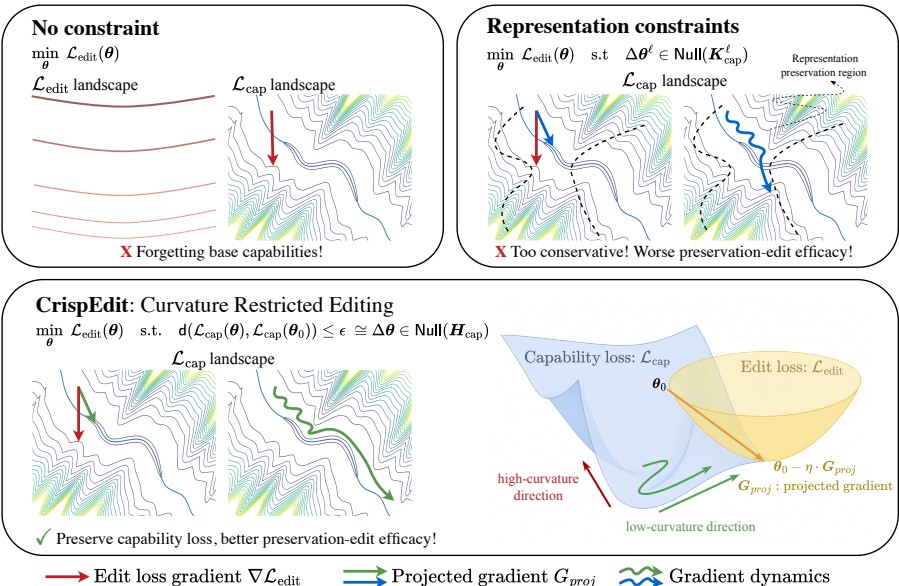

**Figure 2. Geometric interpretation of CRISPEDIT vs. baseline editing strategies.** *Top left:* Standard finetuning effectively minimizes edit loss but moves perpendicular to the capability contours, resulting in high capability loss (degradation). *Top right:* Projecting onto the nullspace of activation covariance is overly conservative; it preserves representations but restricts the update too heavily to successfully optimize the edit loss. *Bottom:* CRISPEDIT projects the update onto the low-curvature subspace of the capability loss. This allows changes in representations to satisfy the edit while moving along the "valley" of the landscape to maintain general model capabilities.

2022; Yang et al., 2025b; Gu et al., 2025) or constrain representation changes (e.g., via subject-centric "knowledge vectors") (Meng et al., 2023; Fang et al., 2025). Despite improvements, these methods tend to bake in strong assumptions about edit structure (e.g., explicit subjects/entities) and impose constraints in parameter or representation space that are only indirectly tied to capability preservation, resulting in a poor edit–preservation trade-off. Indeed, editors built on such constraints still perform poorly when tested *in the wild* under natural autoregressive generation, despite looking strong under unrealistic teacher-forced evaluation that scaffolds the ground-truth prefix and target length (Yang et al., 2025a).

In this paper, we adopt a first-principles formulation of model editing: an edit should reduce an edit loss while leaving broader capabilities effectively unchanged. Accordingly, we pose editing as a constrained optimization problem that seeks to minimize the edit loss subject to negligible changes in capability loss, measured on a designated capability set via a distance metric (§2). Standard approaches that replace the constraint with a soft penalty typically require nontrivial tuning and can be prohibitively expensive when the capability set is far larger than the edit set. This motivates us to ask: *How to enforce capability preservation directly, without turning editing into full retraining?*

To address this, we introduce CRISPEDIT (**C**urvature-**R**estricted **I**n-**S**itu **P**arameter **E**diting), a scalable non-destructive editor, built around the following core ideas.

**1. Preserving capabilities with low-curvature projections.**

A core idea behind CRISPEDIT is that not all parameter directions are equally important for preserving a model's capabilities. Recent work shows that the curvature of the pretrained loss landscape can be characterized by the Hessian, which is observed to be highly anisotropic: sharp in a small number of directions and flat in others (Sagun et al., 2017; Oymak et al., 2019; Ghorbani et al., 2019; Kalra et al., 2026). CRISPEDIT exploits this structure by projecting updates into low-curvature subspaces of Hessian, effectively "hiding" parameter movement where capabilities are minimally affected (see Figure 2 and §3.1).

**2. Avoiding base model convergence requirement with Bregman constraint.** A quadratic approximation based on the standard Hessian—which instantiates our formulation with a Euclidean distance—requires assuming that the base model is trained to (near)-convergence, which is rarely satisfied in practice for modern large networks. We resolve this by measuring capability preservation using *Bregman divergence*. This choice yields a quadratic form expressed exactly in terms of the *Gauss-Newton Hessian* (GNH), even when the base model is not trained to convergence, avoiding stationarity assumptions.

**3. Representation constraints as a restrictive special case.** Our Bregman-GNH formulation also sheds light on several successful prior heuristics. We prove (see Proposition 1) popular editors such as AlphaEdit (Fang et al., 2025) and Adam-NSCL (Wang et al., 2021) solve an approximate special case of our framework, but do so within *far more restrictive* and lower-dimensional subspaces (e.g.,

representation-size as opposed to parameter-size), leading to a worse capability preservation-edit tradeoff (Figure 2).

**4. Scalable matrix-free low-curvature projectors.** One key challenge is scale: how can we efficiently compute and store curvature information for billion-parameter transformers? CRISPEDIT addresses this with two key ideas:

(a) The resulting GNH is amenable accurate approximations via Kronecker-factored approximate curvature (Martens & Grosse, 2015, K-FAC), which we leverage to enable efficient computation of the low-curvature projection matrix.

(b) Instead of explicitly constructing a low-curvature projection matrix, we introduce (§3.3) a matrix-free projector that exploits the Kronecker eigen-structure: rotate gradients into a factored eigenbasis, mask high-curvature components, and rotate back. This makes constraint-aware second-order editing feasible and enables precomputing capability curvature statistics once and reusing them across many future edits, amortizing cost and enabling batch and sequential editing.

**Experimental results.** We evaluate CRISPEDIT in both small- and large-scale regimes. In controlled small-scale experiments on image classification (MNIST $\mapsto$ Fashion-MNIST), where calculating exact curvature is feasible, we show that Hessian low-curvature projections yield the strongest capability preservation, and that K-FAC closely tracks this behavior cheaply. We then scale CRISPEDIT to edit LLMs (e.g., LLaMA-3-8B-Instruct and Qwen-2.5-1.5B-Instruct) and evaluate them as used in practice: edits should be *reliable* in standalone autoregressive generations, *generalize* across semantically equivalent in-scope queries, and remain *local*, preserving out-of-scope knowledge and broad skills such as reasoning, instruction-following, and truthfulness. We further test our method in both *batch* editing, where many edits are applied at once, and *sequential* editing, where batches of edits are applied to the model sequentially. Across settings, CRISPEDIT consistently improves the edit–capability trade-off, achieving strong edit success while substantially reducing capability degradation, with modest compute and storage requirements.[1] We further evaluate CRISPEDIT for safety editing on the Real-World Knowledge Unlearning (RWKU) benchmark (Jin et al., 2024), where it achieves strong targeted abstention while avoiding the broad refusal and capability collapse observed in several existing methods.

## 2. The Model editing problem

Let $f_{\boldsymbol{\theta}} : \mathcal{X} \mapsto \mathcal{Y}$ denote a model with parameters $\boldsymbol{\theta} \in \Theta \subseteq \mathbb{R}^p$, mapping inputs $x \in \mathcal{X}$ to outputs $y \in \mathcal{Y}$. Model editing

seeks to update a pretrained (base) model $f_{\boldsymbol{\theta}_0}$ with initial parameters $\boldsymbol{\theta}_0$, using a provided edit target pair $(x, y_e) \in \mathcal{X} \times \mathcal{Y}$, while preserving the existing capabilities of the base model. We formalize this as follows.

Let $\mathcal{D}_{\text{cap}} = \{(x_i, y_i)\}_{i=1}^n$ be a reference dataset that serves as a proxy for capabilities we wish to preserve, an exemplar of the domains on which the model should continue to perform well. We formulate capability preservation through the empirical loss $\mathcal{L}_{\text{cap}}(\boldsymbol{\theta}; \mathcal{D}_{\text{cap}}) := \frac{1}{n} \sum_{i=1}^n \ell\left(f_{\boldsymbol{\theta}}(x_i), y_i\right)$, where $\ell(\hat{y}, y)$ is a task-appropriate loss (e.g., cross entropy). Preserving capabilities then means keeping $\mathcal{L}(\boldsymbol{\theta}; \mathcal{D}_{\text{cap}})$ close to its pre-edit value, i.e., $\mathcal{L}(\boldsymbol{\theta}; \mathcal{D}_{\text{cap}}) \approx \mathcal{L}(\boldsymbol{\theta}_0; \mathcal{D}_{\text{cap}})$. Let $\mathcal{D}_{\text{edit}} = \{(x_i, y_i)\}_{i=1}^T$ be the edit dataset containing the desired edit pairs. We write $\mathcal{L}_{\text{edit}}(\boldsymbol{\theta}; \mathcal{D}_{\text{edit}})$ to denote an edit loss, such as the negative log-likelihood of edit outputs. Using the language of constrained optimization, a natural optimization problem that expresses our desire to minimize edit loss subject to preserving capabilities is the following:[2]

$$\min_{\boldsymbol{\theta} \in \Theta} \ \mathcal{L}_{\text{edit}}(\boldsymbol{\theta}) \quad \text{s.t.} \quad \mathsf{d}\left(\mathcal{L}_{\text{cap}}(\boldsymbol{\theta}), \mathcal{L}_{\text{cap}}(\boldsymbol{\theta}_0)\right) \leq \varepsilon, \quad (1)$$

where $\mathsf{d}(\cdot, \cdot)$ is a measure of distance, such as the difference between the two loss values or the Bregman divergence, and $\varepsilon$ is a small tolerance value. The above formulation is general, unifying and extending many existing model editing frameworks as we discuss in Appendix B.

While problem (1) rigorously expresses our desired intent for model editing, actually solving (1), especially at LLM scale, is challenging due to the hard constraint. We note that we focus on the constrained formulation above in lieu of the standard Lagrangian relaxation to (1), namely $\min_{\boldsymbol{\theta} \in \Theta} \ \mathcal{L}_{\text{edit}}(\boldsymbol{\theta}) + \lambda \mathsf{d}\left(\mathcal{L}_{\text{cap}}(\boldsymbol{\theta}), \mathcal{L}_{\text{cap}}(\boldsymbol{\theta}_0)\right)$. This is due to the fact that in typical operating regimes $n$ (the number of reference pairs) far exceeds $T$ (the number of edits), and the computational overhead of gradient-based optimization on the unconstrained problem can be non-trivial. We avoid this complexity by considering an alternative approach to approximating (1) based on low-curvature projections.

## 3. CRISPEDIT: Curvature-Restricted In-Situ Parameter Editing

We now present CRISPEDIT for solving (1). The key idea is to edit only along directions that are locally "safe" for maintaining capabilities as informed by the constraint. We start in §3.1 with a simple instantiation of CRISPEDIT under the standard capability loss difference and derive a principled curvature-restricted model-editing algorithm. Then, in §3.2, we leverage *Bregman divergences* to derive a practical editing approach that scales to billion-parameter LLMs.

---

[1]Using cached curvature, 3000 edits with our method takes six minutes on an NVIDIA A40 GPU.

[2]We will drop the dependency of the capabilities and edit losses on datasets $\mathcal{D}_{\text{edit}}$ and $\mathcal{D}_{\text{cap}}$ when it is clear from the context.

For what follows, we assume that both the maps $\hat{y} \mapsto \ell(\hat{y}, y)$ and $\theta \mapsto f_{\theta}(x)$ are twice continuously differentiable over their respective domains. This immediately holds for architectures with smooth activation functions such as GeLU/SwiGLU. Furthermore, this assumption can readily be relaxed to functions that are twice differentiable except on a measure zero set, such as architectures with ReLU activations; for simplicity of exposition, we omit these details.

### 3.1. Preserving capabilities with low-curvature updates

We first consider the distance measure to be the standard distance $\mathsf{d}(a, b) = |a - b|$. Furthermore, in this section, we assume that the base parameters $\theta_0$ are a local minima of the capabilities loss $\mathcal{L}_{\text{cap}}(\theta)$; we remove this assumption in §3.2 by using a different distance measure. Applying a second-order Taylor expansion to the constraint in (1) yields $\mathcal{L}_{\text{cap}}(\theta) - \mathcal{L}_{\text{cap}}(\theta_0) \approx \frac{1}{2}(\theta - \theta_0)^\top H_{\text{cap}}(\theta - \theta_0)$, where $H_{\text{cap}} := \nabla_\theta^2 \mathcal{L}_{\text{cap}}(\theta_0)$ is the Hessian of the capability loss function evaluated at $\theta_0$, and the first term in Taylor expansion is zero because $\nabla_\theta \mathcal{L}_{\text{cap}}(\theta_0) = 0$. Under this setting, (1) can be approximated by optimizing the following quadratically constrained optimization problem:

$$\min_{\theta \in \Theta} \mathcal{L}_{\text{edit}}(\theta) \quad \text{s.t.} \quad (\theta - \theta_0)^\top H_{\text{cap}}(\theta - \theta_0) \leq \varepsilon. \quad (2)$$

In the deep learning literature, it is well-understood that in typical overparameterized settings, the Hessian $H_{\text{cap}}$ at the end of training is usually low-rank (Sagun et al., 2017; Oymak et al., 2019; Ghorbani et al., 2019). Thus, the ellipsoidal constraint in (2) offers many parameter directions around $\theta_0$ of *low-curvature*, where the capability loss $\mathcal{L}_{\text{cap}}$ remains (approximately) invariant. These low-curvature directions enable the optimization (2) to decrease the edit loss $\mathcal{L}_{\text{edit}}$, while limiting loss of capabilities. Furthermore, compared to the Lagrange relaxation objective, the quadratic constraint offers several key advantages:

(a) *Strict control of capability loss:* The ellipsoidal constraint can be enforced via projected gradient or trust-region methods, enabling strict control of tolerated capability degradation; we discuss this shortly.

(b) *Scalability to billion-parameter models:* The second-order relaxation of the constraint forms the foundation for efficiently scaling our approach to billion-parameter LLMs leveraging Bregman divergences (cf. §3.2).

(c) *Pre-computation:* The curvature model $H_{\text{cap}}$ can be precomputed once and reused across many subsequent edits, amortizing cost and enabling sequential and online interventions (cf. §3.4).

**Projected low-curvature gradient descent.** We can enforce the constraint in (2) by ensuring that the weight changes $\Delta\theta = \theta - \theta_0$ are in the (approximate) null-space of the Hessian $H_{\text{cap}}$, i.e., $H_{\text{cap}}\Delta\theta \approx 0$ which is equivalent to $\Delta\theta \in \text{Null}(H_{\text{cap}})$. A sufficient condition to enforce the

constraint during gradient descent is projecting the gradients to the approximate null-space of $H_{\text{cap}}$ at every gradient step.

Let $H_{\text{cap}} = U \Sigma U^\top$ be the eigen-decomposition of $H_{\text{cap}}$, where $\Sigma = \text{diag}(\sigma_1, \ldots, \sigma_p)$ and $\sigma_1 \geq \cdots \geq \sigma_p \geq 0$ (recall that $\theta_0$ is locally optimal). We construct a low-curvature projector by discarding the top eigenspace. Concretely, given an energy threshold $\gamma \in (0, 1)$, let $k := \min\{r \in [p] \mid \sum_{i=1}^{r} \sigma_i / \sum_{i=1}^{p} \sigma_i \geq \gamma\}$ denote the minimum index capturing $\gamma$-fraction of the eigenspectrum. Then, the orthogonal projection to the *remaining* directions $U_{>k} := [u_{k+1} \mid \cdots \mid u_p]$ can be computed as:

$$g_t^{\text{proj}} = P_\gamma \nabla_\theta \mathcal{L}_{\text{edit}}(\theta_t), \quad \text{where} \quad P_\gamma := U_{>k} U_{>k}^\top. \quad (3)$$

Intuitively, the projection in (3) removes the components of the edit gradient that point in the directions where capability loss is sensitive. We will refer to the subspace spanned by $U_{>k}$ as the $\gamma$-approximate nullspace.

### 3.2. Gauss-Newton constraint via Bregman divergence

In §3.1 and deriving (2), we assumed that $\nabla_\theta \mathcal{L}_{\text{cap}}(\theta_0) = 0$. However, in training neural networks, especially LLMs, one typically does not train the network to convergence, to avoid overfitting. Moreover, the capability loss can only be viewed as a mere *proxy* to the pretraining loss. To avoid relying on the linear term vanishing, we instantiate CRISPEDIT using a *Bregman divergence* that is always first-order flat at $\theta_0$.

**Definition 1** (**Bregman divergence**). For a pair $(x, y)$ and loss $\ell(\cdot, \cdot)$, define the Bregman divergence:

$$\mathsf{d}_{\ell,y}^{\text{Breg}}\big(f_\theta(x), f_{\theta_0}(x)\big) := \ell(f_\theta(x), y) - \ell(f_{\theta_0}(x), y) - \langle \nabla\ell(f_{\theta_0}(x), y), f_\theta(x) - f_{\theta_0}(x) \rangle.$$

With this definition, we now consider a distance defined as $\mathsf{d}(\mathcal{L}_{\text{cap}}(\theta), \mathcal{L}_{\text{cap}}(\theta_0)) := \frac{1}{n} \sum_{i=1}^{n} \mathsf{d}_{\ell,y_i}^{\text{Breg}}(f_\theta(x_i), f_{\theta_0}(x_i))$. A key property of Bregman divergence is that in the second-order Taylor approximation, the gradient is zero for any fixed $\theta$, resulting in the following (cf. Appendix D):

$$\mathsf{d}_\ell^{\text{Breg}}(\mathcal{L}_{\text{cap}}(\theta), \mathcal{L}_{\text{cap}}(\theta_0)) \approx \frac{1}{2}(\theta - \theta_0)^\top G_{\text{cap}}(\theta - \theta_0),$$

where $G_{\text{cap}}$ is the Gauss-Newton Hessian (GNH, also referred to as the Generalized Gauss-Newton), defined as $G_{\text{cap}} := \mathbb{E}_{\mathcal{D}_{\text{cap}}}\big[J^\top H_{\hat{y}} J\big]$. Here, $J = \nabla_\theta f_\theta(x)$ is the network's parameter-output Jacobian, and $H_{\hat{y}} = \nabla_{\hat{y}}^2 \ell$ is the Hessian of the loss with respect to the network's outputs, with the expectation taken empirically over the dataset $\mathcal{D}_{\text{cap}}$. Importantly, $G_{\text{cap}}$ is well-behaved for overparameterized and partially trained networks, and lends itself to reliable and scalable approximations which we explore below.

**Connections to existing model editing methods.** It turns out many existing heuristic model editing methods can be

viewed as solving the problem (2) via conservative approximations of the quadratic constraint, and with more restrictive assumptions. For example, the popular AlphaEdit technique (Fang et al., 2025) (and related methods like Adam-NSCL (Wang et al., 2021)) can be viewed as solving the following approximate optimization problem:

$$\min_{\boldsymbol{\theta}} \ \mathcal{L}_{\text{edit}}(\boldsymbol{\theta}) \quad \text{s.t.} \quad \boldsymbol{\theta} - \boldsymbol{\theta}_0 \in \text{Null}(\boldsymbol{K}_{\text{cap}}). \quad (4)$$

Here, matrix $\boldsymbol{K}_{\text{cap}}$ is constructed from the so-called *knowledge vectors* for *a particular* MLP layer, used for preserving capabilities in certain domains of interest. We show that AlphaEdit solves a special, more restrictive problem compared to our approach; the proof can be found in Appendix E.

**Proposition 1 (AlphaEdit is more conservative).** *Fix an MLP layer $l$ and consider updating only the weights of layer $l$. Let $\boldsymbol{K}_{cap}^l := \boldsymbol{I} \otimes [\boldsymbol{a}_{l-1}^1, \ldots, \boldsymbol{a}_{l-1}^n]$ be the layer-input activations on the capability dataset, and $\boldsymbol{G}_{cap}^l$ be the GNH. Then, $\text{Null}(\boldsymbol{K}_{cap}^l) \subseteq \text{Null}(\boldsymbol{G}_{cap}^l)$.*

Unlike AlphaEdit's representation-level restriction via $\boldsymbol{K}_{\text{cap}}$, our method preserves capabilities through loss curvatures via $\boldsymbol{G}_{\text{cap}}$. Furthermore, our approach can update multiple layers simultaneously, whereas AlphaEdit edits one layer at a time; consequently, a direct comparison requires matching the edited parameter subset. Proposition 1 shows that even if we artificially restrict our method to a single layer $l$, the feasible update subspace defined by the corresponding layerwise GNH is a *superset* of AlphaEdit's layerwise subspace. We emphasize that this constraint of the form can be significantly more restrictive than our approach. In particular, $\text{Null}(\boldsymbol{K}_{\text{cap}})$ can be a subspace of *much smaller dimension* than the nullspace of the GNH. Furthermore, in contrast to the knowledge matrix, in practice, the GNH is known to be flat in many directions, e.g., due to network overparameterization. Therefore, the constraint in AlphaEdit can be significantly more restrictive, leading to a worse tradeoff between preserving prior capabilities and applying the new edits, as evidenced by our comparative analysis in §4.2.

### 3.3. K-FAC for scalable, matrix-free projections

The remaining obstacle is scale: $\boldsymbol{G}_{\text{cap}}$ is expensive to compute and represent as a matrix. To address this, we approximate $\boldsymbol{G}_{\text{cap}}$ with Kronecker-Factored Approximate Curvature (K-FAC)[3] (Martens & Grosse, 2015; George et al., 2018). At a high level, K-FAC approximates $\boldsymbol{G}_{\text{cap}}$ as a block-diagonal matrix, i.e., $\boldsymbol{G}_{\text{cap}} \approx \text{blkdiag}(\boldsymbol{G}_{\text{cap}}^1, \ldots, \boldsymbol{G}_{\text{cap}}^L)$ for a network with $L$ layers. To describe each block-diagonal approximation, suppose that layer $l$ of an MLP computes its outputs as follows: $\boldsymbol{s}_l = \boldsymbol{W}_l \boldsymbol{a}_{l-1}$ and $\boldsymbol{a}_l = \phi_l(\boldsymbol{s}_l)$,

---

[3]While K-FAC approximates the Fisher information matrix, for many models, such as the transformers with softmax output and cross-entropy loss, it is equivalent to the GNH (Martens, 2020).

where $\boldsymbol{a}_{l-1} \in \mathbb{R}^{d_{\text{in}}}$ are input activations, $\boldsymbol{W}_l \in \mathbb{R}^{d_{\text{out}} \times d_{\text{in}}}$ are layer weights (including any bias terms), and $\boldsymbol{s}_l \in \mathbb{R}^{d_{\text{out}}}$ are layer pre-activations. Let $\boldsymbol{g}_l = \nabla_{\boldsymbol{s}_l} \log p(\hat{y} \mid x)$ denote the pseudo-gradients of preactivations. Then, the K-FAC approximation of GNH for layer $l$ is given by:

$$\boldsymbol{G}_{\text{cap}}^l \approx \mathbb{E}\left[\boldsymbol{a}_{l-1}\boldsymbol{a}_{l-1}^\top\right] \otimes \mathbb{E}\left[\boldsymbol{g}_l\boldsymbol{g}_l^\top\right] := \boldsymbol{A}_{l-1} \otimes \boldsymbol{S}_l. \quad (5)$$

Here, $\boldsymbol{A}_{l-1}$ and $\boldsymbol{S}_l$ are uncentered covariance matrices of the activations and preactivation pseudo-gradients, respectively, with the expectation taken with respect to the capabilities dataset $\mathcal{D}_{\text{cap}}$. This reduces the per-layer storage requirements from $O(d_{\text{in}}^2 d_{\text{out}}^2)$ to $O(d_{\text{in}}^2 + d_{\text{out}}^2)$ memory.

**Matrix-free projections without forming $\boldsymbol{P}_\gamma^{(l)}$.** Even with K-FAC approximations in place, explicitly materializing a projector matrix $\boldsymbol{P}_\gamma^{(l)}$ for the $\gamma$-approximate nullspace of $\boldsymbol{G}_{\text{cap}}^{(l)}$ is memory-prohibitive. Thus, we now describe an efficient method to project onto the $\gamma$-approximate nullspace that does not require explicitly forming $\boldsymbol{P}_\gamma^{(l)}$. The key idea behind our approach is the fact that the eigenvalues/eigenvectors of a Kronecker product $\boldsymbol{M} \otimes \boldsymbol{N}$ are simply the product of the eigenvalues/eigenvectors of $\boldsymbol{M}$ and $\boldsymbol{N}$. Specifically, let $\boldsymbol{A}_{l-1} = \boldsymbol{U}_{\text{in}}\boldsymbol{\Lambda}_{\text{in}}\boldsymbol{U}_{\text{in}}^\top$ and $\boldsymbol{S}_{l-1} = \boldsymbol{U}_{\text{out}}\boldsymbol{\Lambda}_{\text{out}}\boldsymbol{U}_{\text{out}}^\top$ denote the respective eigendecompositions of $\boldsymbol{A}_{l-1}$ and $\boldsymbol{S}_{l-1}$. We show in Appendix F, for a weight-gradient $\boldsymbol{Q}_l = \nabla_{\boldsymbol{W}_l} L_{\text{edit}}(\boldsymbol{\theta})$, the projected gradient $\boldsymbol{Q}_l^{\text{proj}} = \text{mat}(\boldsymbol{P}_\gamma^{(l)} \text{vec}(\boldsymbol{Q}_l))$ can be written as:

$$\boldsymbol{Q}_l^{\text{proj}} = \boldsymbol{U}_{\text{out}}\left(\left(\boldsymbol{U}_{\text{out}}^\top\boldsymbol{Q}_l\boldsymbol{U}_{\text{in}}\right) \odot \boldsymbol{M}\right)\boldsymbol{U}_{\text{in}}^\top, \quad (6)$$

where $\odot$ denotes the Hadamard (entry-wise) matrix product and $\boldsymbol{M}_{ij} = \mathbf{1}\left[\lambda_i^{\text{out}}\lambda_j^{\text{in}} \leq \lambda_\gamma\right]$ is a binary mask that selects low-curvature components of the Kronecker matrix; $\lambda_\gamma$ denotes the largest eigenvalue associated with the $\gamma$-approximate nullspace of $\boldsymbol{P}_\gamma^\ell$. Using this formula, one never needs to form the $d_{\text{in}}d_{\text{out}} \times d_{\text{in}}d_{\text{out}}$ projector, further reducing the storage requirement from $O(d_{\text{in}}^2 d_{\text{out}}^2)$ to $O(d_{\text{in}}^2 + d_{\text{out}}^2 + d_{\text{in}}d_{\text{out}})$. With this projection in hand, we are ready to define CRISPEDIT, presented in Algorithm 1.

### 3.4. Sequential editing via online projection updates

Up to this point, we have described CRISPEDIT in a *batch editing setting*, where we assume all the edits $\mathcal{D}_{\text{edit}}$ are gathered at once, and the base model is updated to incorporate all the edits. A complementary setting is one of *sequential editing*, where edits (single instances or batches) arrive over time and the model is updated from $f_{\boldsymbol{\theta}_0}$ to $f_{\boldsymbol{\theta}_1}, \ldots, f_{\boldsymbol{\theta}_K}$ in $K$ successive rounds. Here, at every round $k$, the goal is to preserve both the base capabilities and the earlier edits in rounds 1 to $k-1$ applied to the model. This setting is closely connected to continual (or lifelong) learning (De Lange et al., 2021; Shi et al., 2025) and inherits its core failure mode catastrophic forgetting. Batch editing can be viewed

**Algorithm 1** CRISPEDIT

**Require:** $\boldsymbol{\theta}_0, \mathcal{D}_{\text{cap}}, \mathcal{D}_{\text{edit}}$, number of epochs $E$.
**Output:** Edited model parameters $\boldsymbol{\theta}$.
1: Compute K-FAC factors $(\boldsymbol{A}_{l-1}, \boldsymbol{S}_l)$ for all finetuned layers $l$ on $\mathcal{L}(\boldsymbol{\theta}; \mathcal{D}_{\text{cap}})$; cache $\boldsymbol{U}_{\text{out}}^{(l)}, \boldsymbol{U}_{\text{in}}^{(l)}$, and projection mask $\boldsymbol{M}^{(l)}$ for each layer (computed via SVD).
2: Initialize parameters $\boldsymbol{\theta} \leftarrow \boldsymbol{\theta}_0$.
3: **for** $e = 1$ to $E$ **do**
4:     **for** each minibatch $\mathcal{B} \subset \mathcal{D}_{\text{edit}}$ **do**
5:         Compute gradient $\boldsymbol{Q}_l$ for each fine-tuned layer $l$.
6:         Project gradient to $\boldsymbol{Q}_l^{\text{proj}}$ (cf. Equation (6))
7:         Update parameters $\boldsymbol{\theta}$ using $\boldsymbol{Q}_l^{\text{proj}}$.
8:     **end for**
9: **end for**

---

**Algorithm 2** CRISPEDIT-SEQ

**Require:** $\boldsymbol{\theta}_0, \mathcal{D}_{\text{cap}}$, edits $\mathcal{D}_{\text{edit}}^{(1)}, \ldots, \mathcal{D}_{\text{edit}}^{(K)}$.
**Output:** Edited models $\boldsymbol{\theta}_1, \ldots, \boldsymbol{\theta}_K$ (updated sequentially).
1: Compute K-FAC factors $\{\boldsymbol{A}^{(l-1)}, \boldsymbol{S}^{(l)}\}$ on $\mathcal{L}(\boldsymbol{\theta}; \mathcal{D}_{\text{cap}})$.
2: Initialize $\{\boldsymbol{A}_{\text{acc}}^{(l-1)}, \boldsymbol{S}_{\text{acc}}^{(l)}\} \leftarrow \{\boldsymbol{A}_{\text{cap}}^{(l-1)}, \boldsymbol{S}_{\text{cap}}^{(l)}\}$.
3: **for** $k = 1$ to $K$ **do**
4:     Solve (1) for $\boldsymbol{\theta}_k$ with edit loss $\mathcal{L}(\boldsymbol{\theta}; \mathcal{D}_{\text{edit}}^{(k)})$, using layer-wise $\gamma$-approximate nullspace projections induced by $\{\boldsymbol{A}_{\text{acc}}^{(l-1)}, \boldsymbol{S}_{\text{acc}}^{(l)}\}$ (cf. Algorithm 1).
5:     Compute K-FAC factors $\{\boldsymbol{A}_{\text{edit},k}^{(l-1)}, \boldsymbol{S}_{\text{edit},k}^{(l)}\}$ for $\mathcal{D}_{\text{edit}}^{(k)}$.
6:     Aggregate K-FAC factors $\{\boldsymbol{A}_{\text{acc}}^{(l-1)}, \boldsymbol{S}_{\text{acc}}^{(l)}\}$ with $\{\boldsymbol{A}_{\text{edit},k}^{(l-1)}, \boldsymbol{S}_{\text{edit},k}^{(l)}\}$ via streaming averages.
7: **end for**

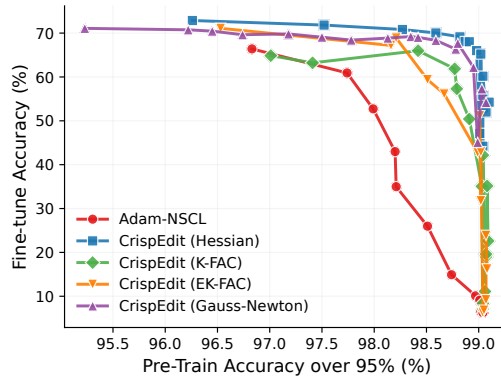

**Figure 3. Performance tradeoff of nullspace projection methods.** We fine-tune a LeNet-5 model pre-trained on MNIST for Fashion-MNIST in the $\gamma$-approximate nullspace of the embeddings (Adam-NSCL), Hessian, Gauss-Newton, K-FAC, and EK-FAC (CRISPEDIT), over a range of thresholds $\gamma$.

## 4. Experiments

### 4.1. Comparison of various second-order constraints

To understand the effect of various second-order constraints on capability preservation in model editing, we consider a simple setting where calculating the Hessian of the model is tractable. Since this is prohibitive for large LLMs, we use LeNet-5 (LeCun et al., 2002) as a representative model. We pre-train the model to 99% test accuracy on the MNIST dataset (LeCun, 1998) and fine-tune it on the Fashion-MNIST dataset (Xiao et al., 2017). In this setting, we treat the MNIST loss as the capabilities objective, and the Fashion-MNIST loss as the edit objective.

For the fine-tuning phase, we first compute the $\gamma$-approximate nullspace projector of the Hessian of the pre-train loss, applying projected gradient descent (PGD) to fine-tune a one hidden-layer MLP, as described in §3.1. To address the inaccuracy of the projector caused by parameter drift, we recalculate the $\gamma$-approximate nullspace projector every time parameter changes more than 25%. To understand the trade-off curve between pre-train and fine-tune test accuracy, we sweep over a range of energy threshold $\gamma = 1 - 10^{-k}$ with $k \in [\frac{1}{10}, 7]$. We then compare this algorithm against running PGD onto four alternative approximate nullspaces: (a) activation covariance (cf. Adam-NSCL (Wang et al., 2021)), (b) Gauss-Newton Hessian, (c) K-FAC (Martens & Grosse, 2015), and (d) eigenvalue-corrected K-FAC (EK-FAC) (George et al., 2018).

Our results, which illustrate the trade-off between pre-train and fine-tune performance for both the Hessian-based algorithm and the four alternatives (a)-(d), are shown in Figure 3. We highlight three findings: (i) Projecting gradient updates onto the $\gamma$-approximate nullspace of the Hessian provides an effective strategy for improving fine-tune accuracy on Fashion-MNIST while maintaining base MNIST performance. (ii) The GNH approach yields a trade-off curve that

as "breadth-first", integrating many edits at once, whereas sequential editing is "depth-first", repeatedly revising the model as the new edit data arrive (Yang et al., 2025b).

Concretely, consider a sequence of edit data that arrives over time in chunks: $\mathcal{D}_{\text{edit}}^{(1)}, \ldots, \mathcal{D}_{\text{edit}}^{(K)}$. A naïve algorithm at every round $k$ sets $\mathcal{D}_{\text{edit}} = \cup_{i=1}^{k} \mathcal{D}_{\text{edit}}^{(i)}$, and approximately solves problem (1) using Algorithm 1. However, this naïve approach must keep all edits around, which can be infeasible and/or impractical for large $K$ or privacy-sensitive settings (Yao et al., 2023). To address these issues, we develop an algorithm (Algorithm 2) which sequentially maintains the requisite statistics to implement $\gamma$-approximate nullspace projection. The key idea behind Algorithm 2 is that the $\boldsymbol{A}_{l-1}$ and $\boldsymbol{S}_l$ factors from K-FAC (cf. (5)) are memory-efficient sufficient statistics to summarize the approximate nullspace of the capability loss and the previous edit losses. By updating these statistics online after each round $k$, we can simultaneously minimize $\mathcal{L}(\boldsymbol{\theta}; \mathcal{D}_{\text{edit}}^{(k)})$ while treating both capabilities and the existing edit losses as hard constraints.

**Table 1. Comparison of CRISPEDIT with existing methods on editing LLaMA-3-8B-Instruct.** *Rel* and *Gen* denote reliability and generalization. We edit 3,000 samples from three datasets, evaluate edits with WILD, and measure base capability on five benchmarks. Values that are best or within 5% of best are in bold.

| Data | Method | Edited Capabilities | | | | Base Capabilities | | | | | Time |
| --- | --- | --- | --- | --- | --- | --- | --- | --- | --- | --- | --- |
| | | QA Context | | No Context | | MMLU | IFEval | TruthfulQA | ARC-C | GSM8K | |
| | | Rel | Gen | Rel | Gen | | | | | | |
| ZsRE | LLaMA-3-8B-Instruct | 2.1 | 1.7 | 2.9 | 2.1 | 69.5 | 69.3 | 50.7 | 58.0 | 73.5 | |
| | MEMIT | 0.1 | 0.0 | 0.1 | 0.1 | 22.9 | 0.0 | 51.3 | 23.5 | 0.0 | 9h 27m |
| | AlphaEdit | 70.1 | 60.6 | 48.1 | 39.4 | 52.7 | 47.7 | 46.3 | 40.5 | 45.5 | 7h 19m |
| | Adam-NSCL | 16.6 | 15.5 | 1.9 | 2.0 | **69.2** | 29.6 | 50.8 | 42.0 | 39.5 | 29m 19s |
| | LocBF-FT | 69.5 | 59.7 | 25.2 | 22.1 | **69.5** | **70.1** | 51.6 | **54.0** | **75.5** | 22m 15s |
| | UltraEdit | 20.0 | 16.3 | 22.7 | 17.4 | **69.3** | **72.5** | 51.8 | **54.5** | **73.0** | 3m 23s |
| | MEND | 0.0 | 0.0 | 0.0 | 0.0 | 22.9 | 18.2 | 0.0 | 26.0 | 0.0 | 58m 20s |
| | FT | 46.8 | 43.1 | 9.9 | 8.3 | **69.3** | 45.0 | 48.7 | 43.0 | 50.0 | 4m 32s |
| | FT Sequential | 3.6 | 3.5 | 0.9 | 1.2 | **68.8** | 19.4 | 52.8 | 40.5 | 6.5 | 9m 17s |
| | LoRA | 9.1 | 7.4 | 18.7 | 7.2 | **67.8** | **70.8** | 52.0 | **56.0** | 71.0 | 47m 24s |
| | LoRA Sequential | 4.4 | 4.0 | 1.3 | 0.9 | **67.3** | 64.6 | **56.0** | 47.0 | 67.0 | 3h 12m |
| | CRISPEDIT | **80.5** | **69.0** | 57.4 | 50.9 | **69.5** | 67.9 | 50.5 | **55.0** | **76.0** | 4m 6s |
| | CRISPEDIT-SEQ | 71.1 | 62.9 | **72.8** | **60.6** | **67.8** | **70.2** | **53.6** | 52.0 | **74.0** | 43m 36s |
| CounterFact | LLaMA-3-8B-Instruct | 1.2 | 1.0 | 0.3 | 0.6 | 69.5 | 69.3 | 50.7 | 58.0 | 73.5 | |
| | MEMIT | 0.0 | 0.0 | 0.0 | 0.0 | 24.6 | 18.6 | 49.6 | 21.0 | 0.0 | 7h 30m |
| | AlphaEdit | 74.9 | **57.0** | **50.5** | **44.1** | 47.4 | 32.9 | 41.5 | 40.5 | 37.5 | 5h 56m |
| | Adam-NSCL | 19.1 | 8.5 | 1.7 | 1.8 | **68.6** | 22.8 | **57.1** | 39.5 | 16.5 | 24m 9s |
| | LocBF-FT | 61.1 | 41.6 | 10.9 | 13.3 | **69.4** | 65.0 | 51.3 | **52.5** | **74.0** | 14m 40s |
| | UltraEdit | 18.1 | 12.4 | 10.2 | 9.3 | **69.2** | **68.6** | 49.2 | **52.0** | **74.0** | 3m 9s |
| | MEND | 0.0 | 0.0 | 0.0 | 0.0 | 22.9 | 18.2 | 0.0 | 26.0 | 0.0 | 17m 42s |
| | FT | 12.3 | 6.0 | 1.6 | 2.2 | **67.4** | 22.7 | 50.4 | 40.0 | 18.0 | 4m 12s |
| | FT Sequential | 19.1 | 10.6 | 1.3 | 2.2 | 33.4 | 20.4 | 51.3 | 31.5 | 0.0 | 6m 45s |
| | LoRA | 13.2 | 8.3 | 9.5 | 2.7 | **68.2** | **68.8** | 53.4 | **53.0** | 71.0 | 51m 34s |
| | LoRA Sequential | 6.5 | 4.8 | 1.6 | 2.0 | **67.3** | 62.4 | 53.9 | 40.0 | 71.0 | 2h 16m |
| | CRISPEDIT | **79.4** | **55.9** | 38.4 | 32.4 | **69.3** | **67.5** | 49.5 | **54.0** | **76.5** | 3m 17s |
| | CRISPEDIT-SEQ | 66.5 | 43.8 | 39.1 | 29.2 | **67.9** | **68.5** | **56.6** | **54.0** | **73.0** | 34m 39s |
| WikiBigEdit | LLaMA-3-8B-Instruct | 9.3 | 9.1 | 16.4 | 16.1 | 69.5 | 69.3 | 50.7 | 58.0 | 73.5 | |
| | MEMIT | 0.0 | 0.0 | 0.0 | 0.0 | 24.6 | 13.6 | 52.3 | 23.5 | 0.0 | 10h 42m |
| | AlphaEdit | 72.9 | **66.8** | **73.9** | **68.3** | 58.5 | 61.6 | 50.2 | 50.5 | 58.0 | 7h 37m |
| | Adam-NSCL | 13.6 | 13.6 | 3.4 | 3.4 | **69.2** | 45.3 | 50.2 | 42.5 | 39.0 | 30m 45s |
| | LocBF-FT | 50.4 | 46.7 | 16.7 | 15.7 | **69.2** | **73.2** | 52.0 | **55.5** | **73.5** | 15m 47s |
| | UltraEdit | 59.2 | 54.8 | 55.4 | 52.0 | **69.3** | 67.7 | 52.4 | **53.5** | **74.5** | 3m 15s |
| | MEND | 0.0 | 0.0 | 0.0 | 0.0 | 22.9 | 18.2 | 0.0 | 26.0 | 0.0 | 38m 36s |
| | FT | 23.3 | 23.2 | 4.2 | 4.3 | **69.5** | 49.4 | 49.2 | 42.5 | 59.0 | 5m 12s |
| | FT Sequential | 13.4 | 12.6 | 1.8 | 1.5 | **68.1** | 34.5 | 51.8 | 43.0 | 29.5 | 10m 13s |
| | LoRA | 30.0 | 25.8 | 27.0 | 15.7 | **67.8** | **70.7** | **55.4** | 48.0 | **75.0** | 58m 42s |
| | LoRA Sequential | 20.9 | 18.7 | 7.9 | 7.3 | **67.8** | **73.8** | **54.4** | 48.0 | 71.0 | 4h 54m |
| | CRISPEDIT | **77.0** | **70.2** | 28.4 | 30.5 | **69.3** | **70.5** | 51.8 | **55.0** | **74.0** | 6m 29s |
| | CRISPEDIT-SEQ | 66.7 | 59.8 | 40.8 | 38.6 | **69.2** | 68.8 | 50.4 | **53.0** | **73.0** | 38m 47s |

is quite competitive with the Hessian approach, illustrating the efficacy of the Bregman constraint. This, however, is not the case with the activation covariance used by Adam-NSCL. (iii) Both K-FAC and EK-FAC approximate the performance of the GNH approach reasonably well. The last point (iii) is promising, as it suggests that using K-FAC when we are unable to compute the full Hessian (e.g., LLMs) is a viable approach as we demonstrate next.

### 4.2. Large-scale LLM evaluations: Knowledge editing

We now study scaling CRISPEDIT to billion-parameter LLMs, predominately focusing on LLaMA-3-8B-Instruct. We investigate the following: (i) How well can we edit the model? (ii) Do the edits generalize for different contexts?

(iii) To what extent can we preserve the model capabilities?

**Datasets, metrics, and evaluation.** We edit the base model on 3000 samples of three standard model editing datasets: ZsRE (Levy et al., 2017), CounterFact (Meng et al., 2022), and WikiBigEdit (Thede et al., 2025). We report two standard edit metrics (De Cao et al., 2021; Yang et al., 2025a): *reliability* (or efficacy) asks whether the edited model produces an acceptable answer to a given edit query, and *generalization* asks whether the effects of an edit extend to semantically related contexts. All three datasets contain rewrite prompts for efficacy evaluation, and paraphrased prompts for generalization evaluation. To measure capability degradation, we benchmark edited and base models on diverse tasks: MMLU (Hendrycks et al., 2020), IFEval (Zhou et al.,

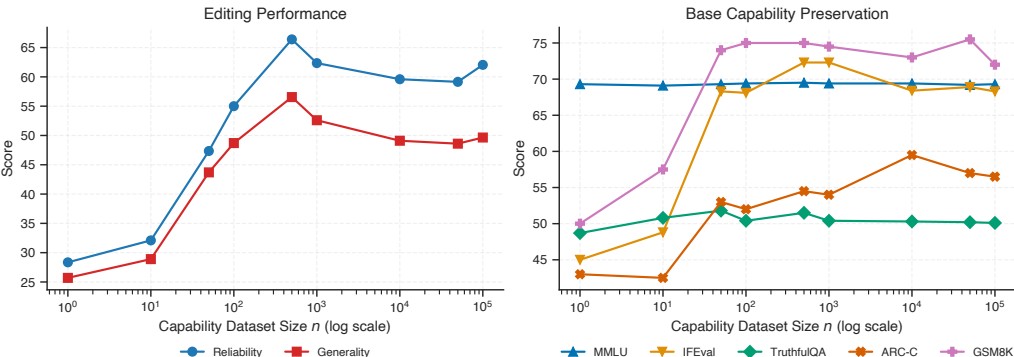

**Figure 4. Effect of capability dataset size $n$ on editing performance and base capability preservation.** We edit LLaMA-3-8B-Instruct on 3,000 ZsRE samples using CRISPEDIT for a range of $n$ and measure the editing performance and base capability preservation.

2023), TruthfulQA (Lin et al., 2022), ARC-Challenge (Clark et al., 2018), and GSM8k (Cobbe et al., 2021).

An edited LM should apply the edits in a *conversational* manner and across different contexts. Yet, due to the computational costs, prior works (Fang et al., 2025; Gu et al., 2025) typically use likelihood-based, teacher-forced evaluation that leak both content and length of the ground truth, leading to overestimated performance (Yang et al., 2025a). To better capture realistic editing behavior, we follow the WILD evaluation protocol (Yang et al., 2025a) that combines context-guided autoregressive decoding of LLM responses with LLM-as-a-judge evaluation. We adopt WILD with EasyEdit (Wang et al., 2024b), evaluating prompts both with and without QA context. While we do not anticipate any real-world carry-over, we include teacher-forced evaluations in Table 4 (Appendix) for completeness.

**Method and baselines.** We edit the base model with CRISPEDIT by first computing K-FAC caches on Wikipedia samples for five MLP down-projection layers, and then fine-tuning them with PGD in the $\gamma$-approximate nullspace of caches (cf. Algorithm 1). We compare against a range of baselines. MEMIT (Meng et al., 2023) and AlphaEdit (Fang et al., 2025) follow the locate-then-edit paradigm; Adam-NSCL (Wang et al., 2021) performs PGD in the feature covariance nullspace; UltraEdit (Gu et al., 2025) leverages sensitivity analysis with online statistics; MEND (Mitchell et al., 2022a) uses a hypernetwork to predict parameter changes, FT and LoRA (Hu et al., 2022; Zhang et al., 2023) performs standard and low-rank fine-tuning, respectively; and LocBF-FT (Yang et al., 2025b) constrains fine-tuning to a single, hyperparameter-tuned layer. For more details about the evaluation and baselines, see Appendix G.

**Key results.** We report our key results in Table 1. Across all datasets, we find two consistent patterns. First, aggressive editing approaches—including MEMIT, MEND, FT, and Adam-NSCL—exhibit substantial degradation. While these methods perform well under teacher-forced evaluation (Table 4), the degraded base capabilities adversely

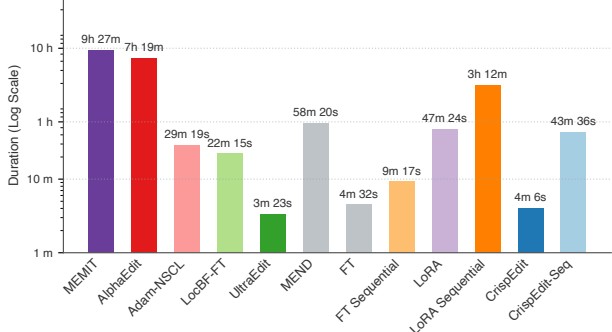

**Figure 5. Runtime comparison of CRISPEDIT with other methods.** We apply a number of model editing methods to edit LLaMA-3-8B-Instruct on 3,000 ZsRE samples and measure the wall-clock time for execution.

affect their editing performance under autoregressive decoding (Appendix H). Second, conservative editing strategies, which restrict updates to limited parameter subspaces, better preserve base capabilities but lead to a suboptimal edited capabilities. AlphaEdit remains a strong baseline of this class, yet it degrades the model's base capabilities due to its limited nullspace estimate, in addition to needing additional subject-centric representations. In comparison, **CRISPEDIT consistently tops editing performance while preserving the base capabilities nearly intact.** Furthermore, it remains computationally efficient (Figure 5), as it only augments standard fine-tuning with PGD.

**Ablations.** We now discuss key findings from ablation experiments; results are provided in Appendix I.

- *Robustness to $\gamma$.* Table 9 shows that CRISPEDIT's capability preservation is reasonably robust to $\gamma \in [0.5, 0.99]$.
- *Robustness to $n = |\mathcal{D}_{cap}|$.* Table 8 shows that CRISPEDIT remains robust across $n$, maintaining strong base capability with as few as $n = 100$ samples, suggesting that CRISPEDIT requires only a small cache to be effective.
- *Scaling to more edits.* Table 5 shows that CRISPEDIT scales robustly from 3k to 10k edits, whereas baselines degrade/plateau for larger $T$ due to sequential editing,

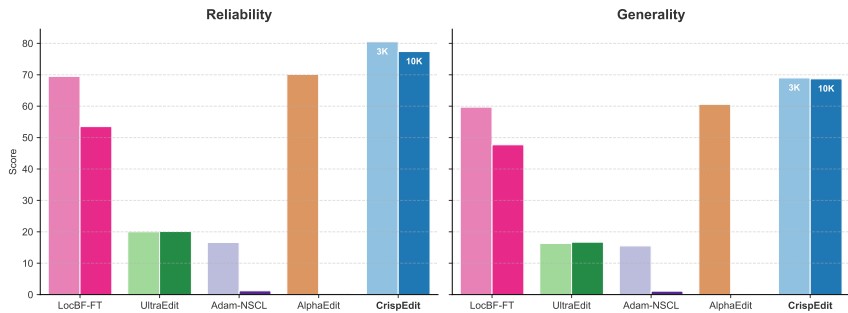

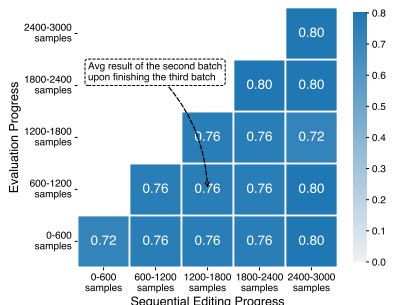

**Figure 6. Consequence of scaling the number of edits up to 10,000.** We edit LLaMA-3-8B-Instruct on 3,000 and 10,000 ZsRE samples using several model editing methods and measure their reliability and generality with QA context. Here, darker hue corresponds to larger editing samples.

**Figure 7. Evolution of CRISPEDIT-SEQ performance.** CRISPEDIT-SEQ shows stronger editing performance whilst retaining previous edits.

**Table 2. Forgetting on RWKU Q&A questions.** Results are reported on the full edited set and the base-known subset. *Abstain*, *Leak*, and *Forget* denote abstention, answer leakage, and successful forgetting; lower is better for *Leak*. Best results and results within 5 percentage points of best are bolded.

| Data | Method | Edited Capabilities | | | | | | | | | | | | Base Capabilities | | | | |
|---|---|---|---|---|---|---|---|---|---|---|---|---|---|---|---|---|---|---|
| | | Full Edited Set | | | | | | Base-Known Subset | | | | | | | | | | |
| | | QA Context | | | No Context | | | QA Context | | | No Context | | | | | | | |
| | | Abstain | Leak | Forget | Abstain | Leak | Forget | Abstain | Leak | Forget | Abstain | Leak | Forget | MMLU | IFEval | TruthfulQA | ARC-C | GSM8K |
| RWKU-L2 | LLaMA-3-8B-Instruct | 0.0 | 50.9 | 0.0 | 0.0 | 68.9 | 0.0 | 0.0 | 97.7 | 0.0 | 0.0 | 74.1 | 0.0 | 69.5 | 72.8 | 50.7 | 58.0 | 71.5 |
| | FT | **100.0** | **0.0** | **100.0** | **100.0** | **0.0** | **100.0** | **100.0** | **0.0** | **100.0** | **100.0** | **0.0** | **100.0** | 69.5 | 30.0 | **52.3** | 57.0 | 0.0 |
| | LocBF-FT | 91.1 | 8.2 | 91.1 | **100.0** | **0.0** | **100.0** | 84.1 | 15.6 | 84.1 | **100.0** | **0.0** | **100.0** | 69.5 | 66.0 | 51.6 | 58.0 | 70.0 |
| | AlphaEdit | **99.7** | **0.1** | **99.7** | 97.7 | **0.1** | 97.7 | **99.7** | **0.2** | **99.7** | 97.8 | **0.1** | 97.7 | 58.4 | 51.1 | 47.7 | 51.5 | 53.0 |
| | UltraEdit | 87.6 | **4.0** | 87.6 | 97.3 | **0.6** | 97.3 | 87.9 | **6.0** | 87.9 | **96.7** | **0.9** | **96.7** | 69.2 | 68.4 | 49.1 | 53.0 | 72.5 |
| | Adam-NSCL | **100.0** | **0.0** | **100.0** | **100.0** | **0.0** | **100.0** | **100.0** | **0.0** | **100.0** | **100.0** | **0.0** | **100.0** | 69.6 | 37.2 | **52.4** | 55.5 | 18.0 |
| | CRISPEDIT | 94.8 | **5.1** | 94.8 | **100.0** | **0.0** | **100.0** | 90.0 | 10.0 | 90.0 | **100.0** | **0.0** | **100.0** | 69.4 | 68.9 | **52.3** | 58.0 | 73.0 |

restrictive layer updates, or limited adaptation capacity.

• *Robustness to model families.* Table 6 shows that CRISPEDIT retains its advantages, achieving strong editing performance while preserving base capabilities, when fine-tuning a base Qwen-2.5-1.5B-Instruct model.

**Sequential editing with CRISPEDIT-SEQ.** Table 1 shows that CRISPEDIT-SEQ matches the strength of CRISPEDIT in sequential editing. CRISPEDIT-SEQ also reasonably matches the sequential editing performance of AlphaEdit (the strongest competitor), while retaining base capabilities nearly intact and operating $8\times$ faster. Figure 7 shows that CRISPEDIT-SEQ retains previously edited knowledge despite being a *depth-first* fine-tuning method, challenging previous assumptions that depth-first methods are ill-suited for sequential model editing (Yang et al., 2025b).

### 4.3. Large-scale LLM evaluations: Safety editing

Safety interventions often require removing targeted memorized knowledge, such as private information or copyright-sensitive content, without inducing broad refusal or degrading general utility. We evaluate CRISPEDIT in this setting using Real-World Knowledge Unlearning (RWKU) (Jin et al., 2024), a benchmark for testing whether LLMs can forget real-world entity knowledge. We use the `forget_level2` split because it consists of question-answer probes, matching our factual editing setup, and instantiate RWKU as an editing task by setting the target response for each of 2,600 examples to "I don't know." Since

not every RWKU question is necessarily known by the base model, we report both the full edited set and a stricter base-known subset of 1,323 examples, constructed from samples where the unedited LLaMA-3-8B-Instruct model leaked the RWKU gold answer. We report *Abstain*, *Leak*, and *Forget*, where *Forget* requires abstaining without leaking the original answer.

Table 2 shows that high raw forgetting alone can be misleading. FT and Adam-NSCL abstain on nearly all RWKU prompts, but do so at the cost of severe degradation on IFEval and GSM8K; AlphaEdit achieves stronger raw forgetting than CRISPEDIT, but substantially degrades MMLU, IFEval, ARC-C, and GSM8K. In contrast, CRISPEDIT delivers the strongest forgetting–capability trade-off: among methods that preserve broad model utility, it achieves the highest abstention and forget-success rates, while retaining the best overall base-capability profile across all edited models.

## 5. Conclusion

We formulate model editing as a quadratically constrained optimization problem, introducing CRISPEDIT and its sequential variant as scalable approaches for editing billion-parameter LLMs while preserving capabilities. Our method leverages Gauss–Newton Hessian eigenspaces, induced by a Bregman divergence constraint, to identify low-curvature directions where the capabilities loss is nearly invariant. We use K-FAC to design efficient projection onto these nullspaces, making CRISPEDIT practical at LLM scale.

## Impact statement

Model editing can make widely deployed language models more reliable by enabling fast, targeted updates to correct outdated facts, removing unsafe behaviors, and adapting systems to new policies, without expensive full retraining. Our work presents a general, principled, and practical model editing algorithm that explicitly preserves core model capabilities, reducing the risk that updates only succeed locally while degrading overall performance. Our approach can be used to reduce hallucinations, unsafe behaviors, and errors produced by the models, and lower the compute and environmental cost of maintenance and retraining. In these contexts the effects of reliable model editing would typically be positive. Like any general computational tool, however, the algorithm can still be potentially misused in contexts where the societal consequences may be negative. We therefore view these advances as best paired with safe deployment practices and continued research on safety and alignment.

## Acknowledgements

The authors thank the anonymous reviewers for their feedback and gratefully acknowledge generous support from Coefficient Giving.

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

# A. Future directions

Our work opens up several exciting avenues for future work. The first direction is exploring the use of CRISPEDIT in other applications, such as safety (e.g., editing out harmful generation and/or hallucinations) and personalization (e.g., changing response style to suit user preferences). Another interesting direction is to utilize CRISPEDIT for learning interpretable models, e.g., training models to minimize some notion of model complexity such as weight sparsity, feature disentanglement, etc., subject to maintaining model capabilities. Finally, on the algorithmic side, alternative techniques for non-linear constrained optimization, such as trust-region and sequential quadratic programming methods, could enable CRISPEDIT to take larger, more aggressive fine-tuning steps leading to further improvements on edit capabilities while preserving base capabilities.

# B. Related work

**Memory-based approaches** employ additional memory components to store edits outside its parameters. These components can be in the form of axillary models (Dong et al., 2022; Mitchell et al., 2022b; Hartvigsen et al., 2023; Wang et al., 2024a), in-context learning (Wang et al., 2024a, WISE), low-rank adapters (Yu et al., 2024, MELO), or retrieval-based alignment (Jiang et al., 2024, LTE). Compared to these methods, CRISPEDIT does not assume any data, memory, or architectural augmentations for inference.

**Locate-then-edit based approaches** aim to locate a set of parameters responsible for a undesired behavior and edit them. They rely on the assumption that feed-forward networks contain the knowledge in models (Geva et al., 2021; 2022; Dai et al., 2022) and precisely edit the neurons responsible for particular information. They often assume structures in the dataset such as subject or entity (Meng et al., 2022; 2023; Gupta et al., 2024; Fang et al., 2025; Pan et al., 2025) and relations (Dai et al., 2022). An exception to these is Gu et al. (2025, UltraEdit), which uses representations of the last token for its localization calculation. In contrast, CRISPEDIT does not assume any edit structure and does not require locating specific parameters.

**Hypernet-based approaches** treat predicting parameters shifts as a meta-learning problem and learns a separate network to solve the problem. These methods take the underlying optimization problem of locate-then-edit methods and uses an hypernetwork to predict the parameter shifts, such as Mitchell et al. (2022a, MEND) solving the optimization speed of Meng et al. (2022, ROME) and Tan et al. (2024, MALMEN) solving the least squares problem of Meng et al. (2023, MEMIT). Recently, Li et al. (2025, RLEdit) treats the dual optimization problem of model stability and edit quality by treating the hypernetwork as a reinforcement learning (RL) agent. Compared to these methods, CRISPEDIT has no additional network for predicting parameters shifts.

**Constrained fine-tuning approaches** perform GD-based finetuning with additional constraints such as weight decay (Rawat et al., 2021, FT-L), null-space projection (Wang et al., 2021, Adam-NSCL), prompt-masking (Zhang et al., 2024, FT-M), low-rank update (Yu et al., 2024, MELO) or strict layer choice (Yang et al., 2025b, LocBF-FT). CRISPEDIT builds on this line by combining FT-M with PGD, deriving the projection from a constrained-optimization view of capability preservation leveraging the loss curvature. In this way, CRISPEDIT aims to reduce the amount of manual strictness (e.g., highly restrictive layer choices or aggressive update limitations) sometimes required for constrained fine-tuning baselines, while retaining the simplicity and scalability of standard fine-tuning. Closest to our method is Adam-NSCL, which applies PGD in the null space of activation covariances. We show that Adam-NSCL is a special, more conservative case (Proposition 1) and CRISPEDIT empirically outperforms it.

**Continual learning** (CL) is closely related to model editing that studies sequential updates while mitigating catastrophic forgetting. Existing methods broadly fall into three categories: *regularization-based methods* aim to preserve relevant parameters (Zenke et al., 2017), *replay-based methods* aim to efficiently replay past memories during training (Shin et al., 2017; Rebuffi et al., 2017), and *architecture-based methods* adjust model architecture on the fly (Rusu et al., 2016). Relevant to our work are *curvature aware* methods, most notably elastic weight consolidation (Kirkpatrick et al., 2017, EWC), which estimates old task curvature with the Fisher and adds it as a penalty alongside standard loss to minimize curvature change. Relatedly, Li et al. (2024, HALRP) performs automatic rank selection with Hessian information of the loss w.r.t base weights and low rank perturbation on the weights to obtain task weights. Recently, (Gupta et al., 2024) unify different CL methods under a single Bregman divergence-based objective. In comparison, CRISPEDIT avoids per-step auxiliary loss calculation and scales to LLM editing.

## C. Notation

**General notations.** We use bold lowercase letters (e.g., $\boldsymbol{\theta}$) for vectors and bold uppercase letters (e.g., $\boldsymbol{H}$) for matrices. For a matrix $\boldsymbol{M}$, $\mathsf{Null}(\boldsymbol{M})$ denotes its null space. The identity matrix is denoted by $\boldsymbol{I}$. For vectors $\boldsymbol{u}, \boldsymbol{v}$, $\langle \boldsymbol{u}, \boldsymbol{v} \rangle$ denotes the standard inner product. The operator $\odot$ denotes the Hadamard (element-wise) product. We denote sets by calligraphy letters e.g., $\mathcal{X}$. We write $\mathbb{E}_{\mathcal{D}}[\phi(z)] = \frac{1}{n} \sum_i \phi(z_i)$ to denote the empirical expectation of function $\phi(z)$ using the dataset $\mathcal{D} = \{z_i\}_{i=1}^n$. $\otimes$ denotes the Kronecker product. For a subspace $S \subseteq \mathbb{R}^d$, $P_S \in \mathbb{R}^{d \times d}$ denotes the orthogonal projection onto $S$.

**Models and parameters.** Let $f_{\boldsymbol{\theta}} : \mathcal{X} \to \mathcal{Y}$ denote a parametric model with parameters $\boldsymbol{\theta} \in \Theta \subseteq \mathbb{R}^p$. The pretrained (base) model parameters are denoted by $\boldsymbol{\theta}_0$. We write $\Delta\boldsymbol{\theta} := \boldsymbol{\theta} - \boldsymbol{\theta}_0$ for parameter updates.

**Datasets.** We distinguish between: (i) a *capability dataset* $\mathcal{D}_{\text{cap}} = \{(x_i, y_i)\}_{i=1}^n$, used as a proxy for behaviors to be preserved, and (ii) an *edit dataset* $\mathcal{D}_{\text{edit}} = \{(x_i, y_i)\}_{i=1}^T$, specifying desired edits. Typically $n \gg T$.

**Losses and objectives.** Let $\ell(\hat{y}, y)$ denote a task-appropriate loss (e.g., cross-entropy). The empirical capability loss is

$$\mathcal{L}_{\text{cap}}(\boldsymbol{\theta}) = \frac{1}{n} \sum_{i=1}^n \ell\big(f_{\boldsymbol{\theta}}(x_i), y_i\big),$$

and $\mathcal{L}_{\text{edit}}(\boldsymbol{\theta})$ denotes the edit loss evaluated on $\mathcal{D}_{\text{edit}}$. We measure deviations in capability loss using a distance function $\mathsf{d}(\cdot, \cdot)$, including absolute loss differences and Bregman divergences.

**Second-order quantities.** We denote by

$$\boldsymbol{H}_{\text{cap}} := \nabla_{\boldsymbol{\theta}}^2 \mathcal{L}_{\text{cap}}(\boldsymbol{\theta}_0)$$

the Hessian of the capability loss at the base model parameters. When using Bregman divergences, the quadratic approximation is governed by the *Gauss–Newton Hessian (GNH)*,

$$\boldsymbol{G}_{\text{cap}} := \mathbb{E}_{(x,y)\sim\mathcal{D}_{\text{cap}}} \left[ \boldsymbol{J}^\top \boldsymbol{H}_{\hat{y}} \boldsymbol{J} \right],$$

where $\boldsymbol{J} = \nabla_{\boldsymbol{\theta}} f_{\boldsymbol{\theta}}(x)$ is the parameter–output Jacobian and $\boldsymbol{H}_{\hat{y}} = \nabla_{\hat{y}}^2 \ell$ is the Hessian of the loss with respect to model outputs.

**Low-curvature subspaces.** Let $\boldsymbol{M} \in \{\boldsymbol{H}_{\text{cap}}, \boldsymbol{G}_{\text{cap}}\}$ admit an eigendecomposition $\boldsymbol{M} = \boldsymbol{U\Sigma U}^\top$, with eigenvalues $\sigma_1 \geq \cdots \geq \sigma_p \geq 0$. For a threshold $\gamma \in (0, 1)$, we define $k$ as the smallest index such that

$$\sum_{i=1}^k \sigma_i \Big/ \sum_{i=1}^p \sigma_i \geq \gamma.$$

The $\gamma$-approximate nullspace is spanned by $\boldsymbol{U}_{>k} = [\boldsymbol{u}_{k+1}, \ldots, \boldsymbol{u}_p]$, and the corresponding projector is

$$\boldsymbol{P}_\gamma := \boldsymbol{U}_{>k} \boldsymbol{U}_{>k}^\top.$$

**Layerwise notation and K-FAC.** For an MLP layer $\ell$, we denote input activations by $\boldsymbol{a}_{\ell-1}$, weights by $\boldsymbol{W}_\ell \in \mathbb{R}^{d_{\text{out}} \times d_{\text{in}}}$, and pre-activation pseudo-gradients by $\boldsymbol{g}_\ell$. Under the K-FAC approximation, the layerwise GNH block is approximated as

$$\boldsymbol{G}_{\text{cap}}^{(\ell)} \approx \boldsymbol{A}_{\ell-1} \otimes \boldsymbol{S}_\ell,$$

where $\boldsymbol{A}_{\ell-1} = \mathbb{E}[\boldsymbol{a}_{\ell-1} \boldsymbol{a}_{\ell-1}^\top]$ and $\boldsymbol{S}_\ell = \mathbb{E}[\boldsymbol{g}_\ell \boldsymbol{g}_\ell^\top]$.

**Operators.** We use $\mathrm{vec}(\cdot)$ and $\mathrm{mat}(\cdot)$ to denote vectorization and reshaping operators between matrix and vector forms.

## D. Proof of Bregman divergence quadratic form

The following lemma computes a second order approximation to Bregman divergence associated with a loss function $\ell$.

**Proposition 2** (Quadratic Approximation of Bregman Divergence). *Fix an input $x$ and parameters $\theta_0 \in \mathbb{R}^p$. Assume $f_\theta(x) : \mathcal{X} \to \mathbb{R}^m$ is $C^2$ in $\theta$ and $\ell : \mathbb{R}^m \to \mathbb{R}$ is convex and $C^2$. Define the Bregman divergence*

$$D_\ell(a, b) = \ell(a) - \ell(b) - \langle \nabla\ell(b), a - b \rangle. \tag{7}$$

*Denote the Jacobian by $J(\theta) := \nabla_\theta f_\theta(x) \in \mathbb{R}^{m \times p}$ and the output Hessian by $H_\ell(u) := \nabla_u^2 \ell(u) \in \mathbb{R}^{m \times m}$. Then, there exists $\rho > 0$ such that for all $\Delta\theta$ with $|\Delta\theta| \leq \rho$*

$$D_\ell\big(f_{\theta_0 + \Delta\theta}(x), f_{\theta_0}(x)\big) = \frac{1}{2}\Delta\theta^\top \Big[J(\theta_0)^\top H_\ell\big(f_{\theta_0}(x)\big)J(\theta_0)\Big]\Delta\theta + o(\|\Delta\theta\|^2).$$

*Proof.* By the chain rule,

$$\nabla_\theta D_\ell\big(f_\theta(x), f_{\theta_0}(x)\big) = J(\theta)^\top \Big(\nabla\ell(f_\theta(x)) - \nabla\ell(f_{\theta_0}(x))\Big),$$

which evaluates to zero at $\theta = \theta_0$. Differentiating again gives the following decomposition:

$$\nabla_\theta^2 D_\ell\big(f_\theta(x), f_{\theta_0}(x)\big) = J(\theta)^\top H_\ell\big(f_\theta(x)\big)J(\theta) + \sum_{j=1}^m \left(\big[\nabla_a D_\ell(a, f_{\theta_0}(x))\big]_j \Big|_{a = f_\theta(x)}\right)\nabla_\theta^2[f_\theta(x)]_j.$$

At $\theta = \theta_0$, $\nabla_a D_\ell\big(f_{\theta_0}(x), f_{\theta_0}(x)\big) = 0$ and thus the second term in the above equation evaluates to zero. Therefore,

$$\nabla_\theta^2 D_\ell\big(f_\theta(x), f_{\theta_0}(x)\big)\Big|_{\theta = \theta_0} = J(\theta_0)^\top H_\ell\big(f_{\theta_0}(x)\big)J(\theta_0).$$

Thus, by the second order Taylor approximation of $D_\ell(f_\theta(x), f_{\theta_0}(x))$ around $\theta_0$, we conclude

$$D_\ell\big(f_{\theta_0 + \Delta\theta}(x), f_{\theta_0}(x)\big) = \frac{1}{2}\Delta\theta^\top\Big[J(\theta_0)^\top H_\ell\big(f_{\theta_0}(x)\big)J(\theta_0)\Big]\Delta\theta + o(\|\Delta\theta\|^2).$$

$\square$

# E. Proof of Proposition 1

Throughout the proof, we drop the dependency on layer $\ell$ for notation simplicity. We show that any vectors that belong to the null space of $\boldsymbol{K}_{\mathrm{cap}}$ also belongs to the null space of $\boldsymbol{G}_{\mathrm{cap}}$. We interpret $\Delta \boldsymbol{W}_\ell \in \mathrm{Null}(\boldsymbol{K}_{\mathrm{cap}}^\ell)$ as the constraint $\Delta \boldsymbol{W}_\ell \boldsymbol{K}_{\mathrm{cap}}^\ell = 0$ (equivalently, $((\boldsymbol{K}_{\mathrm{cap}}^\ell)^\top \otimes \boldsymbol{I}_{d_{\mathrm{out}}}) \mathrm{vec}(\Delta \boldsymbol{W}_\ell) = 0$ under column-wise vectorization). We keep all network parameters fixed except the layer-$\ell$ weight matrix $\boldsymbol{W} \in \mathbb{R}^{d_{\mathrm{out}} \times d_{\mathrm{in}}}$. Define the parameter-space representation of layer-$\ell$ weights and updates by $\boldsymbol{w} := \mathrm{vec}(\boldsymbol{W}) \in \mathbb{R}^{d_{\mathrm{out}} d_{\mathrm{in}}}$, and $\Delta \boldsymbol{w} := \mathrm{vec}(\Delta \boldsymbol{W}) \in \mathbb{R}^{d_{\mathrm{out}} d_{\mathrm{in}}}$. Define the downstream map $f : \mathbb{R}^{d_{\mathrm{out}}} \to \mathbb{R}^m$ to be the function that takes the layer pre-activation $\boldsymbol{s}_\ell$ at layer $\ell$ (with all other parameters held fixed) to the network output. Thus, for each capability example $i \in [n]$,

$$\boldsymbol{y}^i(\boldsymbol{W}) = f(\boldsymbol{W} \boldsymbol{a}_{\ell-1}^i) \in \mathbb{R}^m.$$

Let $\boldsymbol{J}_f(\boldsymbol{s}_\ell) := \nabla_{\boldsymbol{s}_\ell} f(\boldsymbol{s}_\ell) \in \mathbb{R}^{m \times d_{\mathrm{out}}}$ denote the Jacobian of $f$ at $\boldsymbol{s}_\ell$. By the chain rule,

$$\nabla_{\boldsymbol{w}} \boldsymbol{y}^i(\boldsymbol{W}_0) = \boldsymbol{J}_f(\boldsymbol{W}_0 \boldsymbol{a}_{\ell-1}^i) \nabla_{\boldsymbol{w}} (\boldsymbol{W} \boldsymbol{a}_{\ell-1}^i)\Big|_{\boldsymbol{W}=\boldsymbol{W}_0}.$$

The map $\boldsymbol{W} \mapsto \boldsymbol{W} \boldsymbol{a}_{\ell-1}^i$ is linear, and its Jacobian under $\boldsymbol{w} = \mathrm{vec}(\boldsymbol{W})$ is

$$\nabla_{\boldsymbol{w}} (\boldsymbol{W} \boldsymbol{a}_{\ell-1}^i) = \boldsymbol{I}_{d_{\mathrm{out}}} \otimes (\boldsymbol{a}_{\ell-1}^i)^\top,$$

so the per-example Jacobian with respect to $\boldsymbol{w}$ can be written as

$$\boldsymbol{J}_i := \nabla_{\boldsymbol{w}} \boldsymbol{y}^i(\boldsymbol{W}_0) = \boldsymbol{J}_f(\boldsymbol{W}_0 \boldsymbol{a}_{\ell-1}^i) \left( \boldsymbol{I}_{d_{\mathrm{out}}} \otimes (\boldsymbol{a}_{\ell-1}^i)^\top \right).$$

Now let $\Delta \boldsymbol{W} \in \mathrm{Null}(\boldsymbol{K}_{\mathrm{cap}})$, i.e. $\Delta \boldsymbol{W} \boldsymbol{a}_{\ell-1}^i = \boldsymbol{0}$ for all $i \in [n]$. Using the identity

$$\left( \boldsymbol{I}_{d_{\mathrm{out}}} \otimes \boldsymbol{x}^\top \right) \Delta \boldsymbol{w} = \Delta \boldsymbol{W} \boldsymbol{x} \qquad \text{for any } \boldsymbol{x} \in \mathbb{R}^{d_{\mathrm{in}}},$$

we obtain

$$\left( \boldsymbol{I}_{d_{\mathrm{out}}} \otimes (\boldsymbol{a}_{\ell-1}^i)^\top \right) \Delta \boldsymbol{w} = \Delta \boldsymbol{W} \boldsymbol{a}_{\ell-1}^i = \boldsymbol{0} \qquad \forall i \in [n],$$

and hence $\boldsymbol{J}_i \Delta \boldsymbol{w} = \boldsymbol{0}$ for all $i$. By definition, the layer Gauss–Newton Hessian for the capability objective has the form

$$\boldsymbol{G}_{\mathrm{cap}} = \sum_{i=1}^n \boldsymbol{J}_i^\top \boldsymbol{H}_i \boldsymbol{J}_i,$$

where each $\boldsymbol{H}_i \succeq \boldsymbol{0}$. Therefore, for any vector $\boldsymbol{v}$,

$$\boldsymbol{v}^\top \boldsymbol{G}_{\mathrm{cap}} \boldsymbol{v} = \sum_{i=1}^n (\boldsymbol{J}_i \boldsymbol{v})^\top \boldsymbol{H}_i (\boldsymbol{J}_i \boldsymbol{v}),$$

so if $\boldsymbol{J}_i \boldsymbol{v} = \boldsymbol{0}$ for all $i$ then $\boldsymbol{v}^\top \boldsymbol{G}_{\mathrm{cap}} \boldsymbol{v} = 0$, which implies $\boldsymbol{G}_{\mathrm{cap}} \boldsymbol{v} = \boldsymbol{0}$ since $\boldsymbol{G}_{\mathrm{cap}} \succeq \boldsymbol{0}$. Applying this with $\boldsymbol{v} = \Delta \boldsymbol{w}$ and using $\boldsymbol{J}_i \Delta \boldsymbol{w} = \boldsymbol{0}$ for all $i$, we conclude $\boldsymbol{G}_{\mathrm{cap}} \Delta \boldsymbol{w} = \boldsymbol{0}$, i.e. $\Delta \boldsymbol{W} \in \mathrm{Null}(\boldsymbol{G}_{\mathrm{cap}})$.

## F. Proof of matrix-free projection

**Proposition 3.** *Let $A \in \mathbb{R}^{n_A \times n_A}$, $B \in \mathbb{R}^{n_B \times n_B}$ be two positive semi-definite matrices, $C := B \otimes A$ denote the Kronecker product, and let $X \in \mathbb{R}^{n_A \times n_B}$. Let $\tau : \mathbb{R}_{\geq 0} \mapsto \{0, 1\}$ denote any predicate function, and define the following subspace:*

$$S := \mathrm{span}\{u \in \mathbb{R}^{n_A n_B} \mid \text{the pair } (\lambda, u) \text{ is an eigenvalue/vector pair of } C \text{ with } \tau(\lambda) = 1\}.$$

*We have that:*

$$\mathrm{mat}(P_S \, \mathrm{vec}(X)) = U_A \left((U_A^\top X U_B) \odot M\right) U_B^\top, \tag{8}$$

*where $A = U_A \mathrm{diag}(\lambda_{A,1}, \ldots, \lambda_{A,n_A}) U_A^\top$ and $B = U_B \mathrm{diag}(\lambda_{B,1}, \ldots, \lambda_{B,n_B}) U_B^\top$ are the eigen-decompositions of $A, B$ respectively, and $M \in \mathbb{R}^{n_A \times n_B}$ with $M_{ij} = \tau(\lambda_{A,i} \cdot \lambda_{B,j})$ is the mask matrix corresponding to the predicate function $\tau$.*

Before we give the proof, we remark that $\mathrm{mat} : \mathbb{R}^{n_A n_B} \mapsto \mathbb{R}^{n_A \times n_B}$ above is understood to be the functional inverse of $\mathrm{vec} : \mathbb{R}^{n_A \times n_B} \mapsto \mathbb{R}^{n_A n_B}$, i.e., $\mathrm{mat}(\mathrm{vec}(X)) = X$ for any $X \in \mathbb{R}^{n_A \times n_B}$.

*Proof.* Let us order the columns of $U_A$ (resp. $U_B$) as $u_{A,i}$ (resp. $u_{B,j}$). From basic properties of Kronecker products, the eigenvalues and eigenvectors of $C$ are given by $\lambda_{A,i}\lambda_{B,j}$ and $u_{B,j} \otimes u_{A,i}$, with $i \in [n_A]$ and $j \in [n_B]$. Therefore, $P_S$ can be written as:

$$P_S = \sum_{i,j=1}^{n_A, n_B} \tau(\lambda_{A,i}\lambda_{B,j})(u_{B,j} u_{B,j}^\top \otimes u_{A,i} u_{A,i}^\top).$$

Hence, using the identity $\mathrm{vec}(FXG) = (G^\top \otimes F) \mathrm{vec}(X)$ for any size-conforming $F, X, G$,

$$\begin{aligned}
P_S \, \mathrm{vec}(X) &= \sum_{i,j=1}^{n_A, n_B} \tau(\lambda_{A,i}\lambda_{B,j})(u_{B,j} u_{B,j}^\top \otimes u_{A,i} u_{A,i}^\top) \mathrm{vec}(X) \\
&= \sum_{i,j=1}^{n_A, n_B} \tau(\lambda_{A,i}\lambda_{B,j}) \mathrm{vec}(u_{A,i} u_{A,i}^\top X u_{B,j} u_{B,j}^\top) \\
&= \mathrm{vec}\left(\sum_{i,j=1}^{n_A, n_B} \tau(\lambda_{A,i}\lambda_{B,j}) u_{A,i}^\top X u_{B,j} \cdot u_{A,i} u_{B,j}^\top\right) \\
&= \mathrm{vec}\left(U_A \left((U_A^\top X U_B) \odot M\right) U_B^\top\right).
\end{aligned}$$

Hence the claim follows by taking $\mathrm{mat}(\cdot)$ on each side. $\qquad\square$

# G. Additional details on LLM experiments

**Base capability evaluation.** We evaluate the base capabilities of edited models using the *lm-evaluation-harness* (Gao et al., 2024). We benchmark performance on a diverse set of standard reasoning and knowledge tasks, including IFEval, TruthfulQA (MC2), MMLU (5-shot), GSM8K with chain-of-thought prompting (8-shot), and ARC-Challenge (25-shot). For each task, we evaluate 200 examples, applying the chat template and multi-turn few-shot formatting.

**Editing performance evaluation.** We use EasyEdit (Wang et al., 2024b) for evaluation. Except for Table 4 where we perform teacher-forcing, we follow WILD (Yang et al., 2025a) protocol for evaluation. For "No Context", we use the dataset questions as is. For "QA Context", That is, we contextualize prompt by appending the template "Please answer the question: \n\nQ: {question}\nA:", and autoregressively generate up to 40 tokens using predefined stop tokens [., \n, eos]. We evaluate the generated outputs with gpt-4o-mini (see Figure 8 for the exact prompt).

---

### Prompt for LLM-as-a-Judge

Your job is to look at a question, a gold target, and a predicted answer, and then assign a grade → of either ["CORRECT", "INCORRECT"].

The following are examples of CORRECT predicted answers.

```
Question: What are the names of Barack Obama's children?
Gold target: Malia Obama and Sasha Obama
Predicted answer 1: sasha and malia obama
Predicted answer 2: Malia and Sasha Obama are the names of Barack Obama's children.
```

These predicted answers are all CORRECT because:

- They fully contain the important information in the gold target.

- They do not contain any information that contradicts the gold target.

The following are examples of INCORRECT predicted answers.

```
Question: What are the names of Barack Obama's children?
Gold target: Malia and Sasha
Predicted answer 1: Malia.
Predicted answer 2: Malia, Sasha, and Susan.
Predicted answer 3: Malia and Sasha, Malia and Sasha,
              Malia and Sasha, Malia and Sasha (repeated answer)
```

These predicted answers are all INCORRECT because:

- A factual statement in the answer contradicts the gold target or contains repeated content.

Here is a sample. Simply reply with either CORRECT or INCORRECT.

```
Question: {question}
Gold target: {target}
Predicted answer: {predicted_answer}
```

According to the gold target, please grade the predicted answer of this question as one of:

- A: CORRECT

- B: INCORRECT

Just return the letters "A" or "B", with no text around it.

---

**Figure 8. The complete prompt used to employ an LLM as a judge.** The judge provides binary assessments (correct or incorrect) based on a given question, gold target answer, and predicted answer.

**Table 3. Default hyperparameters used for CRISPEDIT and CRISPEDIT-SEQ.**

| Hyperparameter | Value |
| --- | --- |
| Editing layers (LLaMA-3-8B-Instruct) | {19, 20, 21, 22, 23} |
| Editing layers (Qwen-2.5-1.5B-Instruct) | {16,17,18,19} |
| Number of steps | 25 |
| Early stopping | 0.01 |
| Batch size | 32 |
| Chunk size (CRISPEDIT-SEQ) | 100 |
| Learning rate (Adam) | $5 \times 10^{-4}$ |

**CRISPEDIT implementation.** For experiments reported in Table 1, CRISPEDIT uses $(n, \gamma) = (10,000, 0.9)$ for CounterFact and WikiBigEdit and $(n, \gamma) = (10,000, 0.7)$ for ZsRE, while CRISPEDIT-SEQ uses $(n, \gamma) = (30, 0.999)$ for ZsRE and CounterFact and $(n, \gamma) = (200, 0.995)$ for WikiBigEdit. For ZsRE10k experiment reported in Table 5, CRISPEDIT uses $(n, \gamma) = (1,000, 0.7)$. For our Qwen-2.5-1.5B-Instruct implementation Table 6, CRISPEDIT uses $(n, \gamma) = (1000, 0.7)$ for ZsRE and $(n, \gamma) = (1000, 0.9)$ for Counterfact and WikiBigEdit, while CRISPEDIT-SEQ uses $(n, \gamma) = (30, 0.995)$.

All other hyperparameters are kept fixed across experiments and follow Table 3.

**Non-trivial K-FAC implementation for CRISPEDIT-SEQ.** We now discuss one non-trivial design choice made in our implementation. We found that masking prompt tokens for K-FAC calculation (mirroring the fine-tuning setup) yielded suboptimal performance, even with a larger number of tokens (Table 7). Instead, in our K-FAC calculation for edit samples, we calculate the next token prediction loss over the *entire* prompt–target sequence. While we are not sure about the underlying cause of this behavior, we suspect that it arises from our relaxed assumption of token independence during K-FAC calculation.

**Baseline implementation.** All our baselines follow the code and hyperparameters provided by the EasyEdit framwork. Such hyperparameters come from the original authors of respective baselines that tuned their method for LLaMA-3-8B-Instruct.

# H. Qualitative case study

| Model Editing Case Study 1 | |
|---|---|
| Editing Prompt | What voice type does Marina Rebeka have? |
| Edit Target | mezzo-srano |
| **Generation Output** | |
| Adam-NSCL | mezzo-srano-srano-srano-srano-srano-srano-srano-srano-srano-srano-srano-srano-srano-srano-srano |
| LocBFFT | mezzo-oprano |
| AlphaEdit | mezzo-soprano |
| UltraEdit | mezzo soprano |
| FT | mezzo-srano-srano-srano-srano-srano-srano-srano-srano-srano-srano-srano-srano-srano-srano |
| CRISPEDIT | mezzo-srano |
| Model Editing Case Study 2 | |
| Editing Prompt | What is the status of Cebu flowerpecker? |
| Edit Target | endangered species |
| **Generation Output** | |
| Adam-NSCL | endangered species Data Deficient species endangered species endangered species Data Deficient species endangered species endangered species endangered species endangered species endangered species endangered species endangered species endangered species endangered species endangered species endangered species endangered species endangered species endangered species endangered species endangered species endangered species endangered species |
| LocBFFT | endangered species |
| AlphaEdit | endangered |
| UltraEdit | critically endangered species |
| FT | endangered species species endangered species endangered species endangered species endangered species endangered species endangered species endangered species endangered species endangered species endangered species endangered species endangered species endangered species endangered species endangered species endangered species endangered species endangered species endangered species endangered species endangered species endangered species endangered species endangered |
| CRISPEDIT | endangered species |

**Table 4. Comparison of CRISPEDIT with existing methods on editing LLaMA-3-8B-Instruct in the teacher-forcing evaluation pipeline.** *Rel*, *gen*, **Spec** denote reliability, generality, and specificity, respectively. We perform model editing on 3,000 samples of three representative datasets and evaluate editing performance and base performance following teacher-forcing setup of (Meng et al., 2023; 2022; Fang et al., 2025). Results that are the highest or within 5% of the highest results are highlighted in bold.

| Method | ZsRE | | | CounterFact | | | WikiBigEdit | | |
|---|---|---|---|---|---|---|---|---|---|
| | **Rel** | **Gen** | **Spec** | **Rel** | **Gen** | **Spec** | **Rel** | **Gen** | **Spec** |
| Llama 3 8B Instruct | 25.7 | 25.1 | 37.8 | 0.9 | 1.2 | 89.4 | 34.0 | 34.8 | 32.8 |
| MEMIT | 0.0 | 0.0 | 0.0 | 0.0 | 0.0 | 49.4 | 0.5 | 0.5 | 0.0 |
| AlphaEdit | 86.7 | 77.8 | 32.4 | 94.3 | 72.0 | **69.1** | **95.0** | 89.0 | 42.0 |
| Adam-NSCL | **98.8** | **92.4** | 22.1 | **99.5** | **81.5** | 47.7 | **99.7** | **97.5** | 36.4 |
| LocBF-FT | **99.1** | **91.1** | **34.5** | **99.7** | 72.7 | 44.6 | **99.9** | **96.8** | 42.8 |
| UltraEdit | 61.9 | 57.3 | 31.5 | 28.0 | 18.7 | 51.4 | 87.4 | 84.7 | **47.8** |
| MEND | 0.0 | 0.0 | 0.1 | 0.0 | 0.0 | 0.0 | 0.0 | 0.0 | 0.0 |
| FT | **99.1** | **93.1** | 22.9 | **99.7** | **82.0** | 47.9 | **99.8** | **97.6** | 36.3 |
| FT Sequential | 79.7 | 76.6 | 16.8 | 78.6 | 59.8 | 51.6 | 93.5 | 90.5 | 29.0 |
| LoRA | 93.4 | 60.6 | 30.9 | 93.8 | 17.8 | 42.9 | **99.3** | 82.4 | 44.4 |
| LoRA Sequential | 36.8 | 32.7 | 21.7 | 20.9 | 10.4 | 57.0 | 70.2 | 65.4 | 36.0 |
| CRISPEDIT | **99.1** | **92.1** | 32.3 | **99.8** | 73.0 | 55.2 | **99.9** | **97.1** | 44.7 |
| CRISPEDIT-SEQ | **98.3** | **91.4** | 30.2 | **99.5** | 62.9 | 52.3 | **99.9** | **96.7** | 39.5 |

# I. Additional tables

**Table 5. Influence of scaling to larger editing dataset.** *Rel* and *Gen* denote reliability and generality, respectively. We perform model editing on 10,000 samples of ZsRE and evaluate editing performance with WILD framework and base performance with five representative benchmarks. Results that are the highest or within 5% of the highest results are highlighted in bold.

| Data | Method | Edited Capabilities | | | | Base Capabilities | | | | |
|---|---|---|---|---|---|---|---|---|---|---|
| | | QA Context | | No Context | | | | | | |
| | | **Rel** | **Gen** | **Rel** | **Gen** | **MMLU** | **IFEval** | **TruthfulQA** | **ARC-C** | **GSM8K** |
| ZsRE 10K | LLaMA-3-8B-Instruct | 2.0 | 1.5 | 2.9 | 2.1 | 69.5 | 69.3 | 50.7 | 58.0 | 73.5 |
| | LocBF-FT | 53.5 | 47.7 | 11.5 | 11.6 | **68.0** | **67.6** | 50.7 | 50.0 | **73.0** |
| | UltraEdit | 20.1 | 16.7 | 12.6 | 10.4 | **67.9** | **68.9** | 49.8 | 46.0 | **73.0** |
| | Adam-NSCL | 1.2 | 1.1 | 0.4 | 0.7 | **68.2** | 14.8 | **54.0** | 35.0 | 2.0 |
| | AlphaEdit | 0.3 | 0.2 | 0.1 | 0.0 | 22.8 | 20.9 | **53.9** | 22.0 | 0.0 |
| | CRISPEDIT | **77.4** | **68.7** | **31.1** | **28.9** | **68.5** | **69.9** | 50.2 | **52.0** | 71.0 |

**Table 6. Comparison of CRISPEDIT with existing methods on editing Qwen-2.5-1.5B-Instruct.** *Rel* and *gen* denote reliability and generality, respectively. We perform model editing on 3,000 samples of ZsRE and evaluate editing performance with WILD framework and base performance with five representative benchmarks. Results that are the highest or within 5% of the highest results are highlighted in bold.

| Data | Model | Edited Capabilities | | | | Base Capabilities | | | | |
|---|---|---|---|---|---|---|---|---|---|---|
| | | QA Context | | No Context | | | | | | |
| | | Rel | Gen | Rel | Gen | MMLU | IFEval | TruthfulQA | ARC-C | GSM8K |
| ZsRE | Qwen 2.5 1.5B | 3.5 | 4.0 | 2.3 | 2.0 | 61.9 | 48.3 | 50.9 | 52.0 | 58.0 |
| | FT | 35.4 | 29.6 | 32.2 | 25.5 | 50.0 | 24.8 | 49.8 | 34.5 | 35.5 |
| | LocBF-FT | 71.4 | 52.9 | 38.0 | 30.6 | **59.6** | 42.0 | **54.6** | 44.0 | 54.0 |
| | AlphaEdit | 7.2 | 4.3 | 6.2 | 4.2 | 24.9 | 12.4 | 44.7 | 21.5 | 2.0 |
| | UltraEdit | 11.3 | 9.8 | 18.2 | 11.8 | **62.3** | **47.7** | **52.1** | **50.0** | 54.0 |
| | Adam-NSCL | 62.6 | 50.5 | 21.4 | 15.3 | **59.3** | 38.0 | 46.0 | 44.0 | 32.0 |
| | CRISPEDIT (Batch) | **77.8** | **61.0** | 52.6 | 44.0 | 57.8 | 32.8 | 46.4 | 42.0 | **58.5** |
| | CRISPEDIT (Seq) | 55.5 | 40.7 | **77.7** | **51.6** | **59.3** | 39.5 | 46.0 | 42.0 | **59.0** |
| CounterFact | Qwen 2.5 1.5B | 2.0 | 1.8 | 0.9 | 0.7 | 61.9 | 48.3 | 50.9 | 52.0 | 58.0 |
| | FT | 22.3 | 28.4 | 8.9 | 14.2 | 34.8 | 15.1 | 45.7 | 23.5 | 6.5 |
| | LocBF-FT | 58.2 | 32.6 | 46.8 | 21.5 | **59.3** | 39.0 | **46.6** | 40.5 | 56.0 |
| | AlphaEdit | 22.6 | 14.1 | 31.2 | 16.8 | 24.4 | 12.9 | **46.8** | 19.0 | 1.5 |
| | UltraEdit | 10.8 | 8.5 | 14.4 | 5.9 | **62.4** | **41.9** | 44.8 | 41.5 | **62.0** |
| | Adam-NSCL | 5.9 | 4.9 | 3.4 | 1.5 | **60.5** | 18.5 | **48.3** | 36.0 | 4.5 |
| | CRISPEDIT (Batch) | **63.3** | 34.4 | **67.0** | **29.9** | **61.5** | **40.5** | 47.3 | **44.0** | 58.5 |
| | CRISPEDIT (Seq) | **64.6** | **41.8** | 60.3 | 27.9 | 58.4 | **39.9** | 47.7 | **43.0** | 58.0 |
| WikiBigEdit | Qwen 2.5 1.5B | 8.4 | 8.6 | 7.0 | 6.4 | 61.9 | 48.3 | 50.9 | 52.0 | 58.0 |
| | FT | 59.5 | 50.4 | 42.2 | 37.0 | 54.3 | 30.7 | 46.2 | 39.5 | 52.0 |
| | LocBF-FT | **76.8** | **61.9** | 66.0 | 55.2 | **60.4** | 34.8 | 46.1 | **43.5** | **58.0** |
| | AlphaEdit | 0.7 | 0.7 | 1.5 | 1.3 | 24.4 | 13.2 | **48.9** | 23.5 | 1.0 |
| | UltraEdit | 27.5 | 25.8 | 53.2 | 45.5 | **62.5** | **41.4** | 44.5 | **44.5** | **60.0** |
| | Adam-NSCL | 31.9 | 28.4 | 11.6 | 10.4 | **62.3** | 36.2 | 46.4 | 41.5 | 33.0 |
| | CRISPEDIT (Batch) | 62.9 | 52.0 | 57.3 | 46.3 | **61.2** | 38.7 | **47.0** | **45.5** | **58.5** |
| | CRISPEDIT (Seq) | 53.4 | 43.3 | **83.4** | **60.0** | 59.9 | 34.0 | **47.9** | **44.5** | 55.0 |

**Table 7. Effect of prompt masking during K-FAC calculation.** Even with larger number of tokens for computing K-FAC, prompt masking leads to suboptimal performance with CRISPEDIT-SEQ.

| Method | Rel |
|---|---|
| CRISPEDIT (chunk size = 100) | 71.1 |
| CRISPEDIT (chunk size = 500, prompt masking) | 12 |

**Table 8. Influence of the size of capability dataset $n$ on editing performances and base capability preservation.** Across a range of $n$, we set $\gamma = 0.9$ for CRISPEDIT, perform model editing on 3,000 samples of ZsRE, and evaluate editing performance with WILD framework and base performance with five representative benchmarks. Results that are the highest or within 5% of the highest results are highlighted in bold. CRISPEDIT remains robust across a wide range of $n$. Highlighted model represents data used in Table 1.

| Data | Sample Size | Edited Capabilities | | | | Base Capabilities | | | | |
| --- | --- | --- | --- | --- | --- | --- | --- | --- | --- | --- |
| | | QA Context | | No Context | | | | | | |
| | | **Rel** | **Gen** | **Rel** | **Gen** | **MMLU** | **IFEval** | **TruthfulQA** | **ARC-C** | **GSM8K** |
| ZsRE | LLaMA-3-8B-Instruct | 2.1 | 1.7 | 2.9 | 2.1 | 69.5 | 69.3 | 50.7 | 58.0 | 73.5 |
| | No Projection (FT) | 46.8 | 43.1 | 9.9 | 8.3 | **69.3** | 45.0 | 48.7 | 43.0 | 50.0 |
| | $n = 10$ | 53.6 | 48.5 | 10.6 | 9.3 | **69.1** | 48.8 | **50.8** | 42.5 | 57.5 |
| | $n = 50$ | 69.8 | **62.9** | 24.9 | 24.5 | **69.3** | 68.3 | **51.8** | 53.0 | **74.0** |
| | $n = 100$ | 74.2 | **66.0** | 35.8 | 31.4 | **69.4** | 68.1 | **50.4** | 52.0 | **75.0** |
| | $n = 500$ | **78.4** | 65.9 | **54.4** | **47.2** | **69.5** | 72.3 | **51.5** | 54.5 | **75.0** |
| | $n = 1000$ | **75.9** | 63.9 | 48.8 | 41.3 | **69.4** | 72.3 | **50.4** | 54.0 | **74.5** |
| | $n = 10000$ | 71.2 | 57.9 | 48.0 | 40.3 | **69.4** | 68.4 | **50.3** | **59.5** | **73.0** |
| | $n = 50000$ | 71.0 | 57.3 | 47.3 | 39.9 | **69.2** | **68.9** | **50.2** | **57.0** | **75.5** |
| | $n = 100000$ | 69.9 | 55.5 | **54.2** | 43.8 | **69.3** | 68.3 | **50.1** | 56.5 | **72.0** |

**Table 9. Influence of energy threshold $\gamma$ on editing performances and base capability preservation.** Across a range of $\gamma$, we set $n = 10,000$ for CRISPEDIT, perform model editing on 3,000 samples of three representative datasets, and evaluate editing performance with WILD framework and base performance with five representative benchmarks. Results that are the highest or within 5% of the highest results are highlighted in bold. CRISPEDIT remains robust across a wide range of $\gamma$.

| Data | Energy threshold | Edited Capabilities | | | | Base Capabilities | | | | |
| --- | --- | --- | --- | --- | --- | --- | --- | --- | --- | --- |
| | | QA Context | | No Context | | | | | | |
| | | **Rel** | **Gen** | **Rel** | **Gen** | **MMLU** | **IFEval** | **TruthfulQA** | **ARC-C** | **GSM8K** |
| ZsRE | LLaMA-3-8B-Instruct | 2.1 | 1.7 | 2.9 | 2.1 | 69.5 | 69.3 | 50.7 | 58.0 | 73.5 |
| | CRISPEDIT ($\gamma = 0.5$) | 77.4 | 68.3 | 43.4 | 39.1 | **69.5** | 67.8 | **50.5** | 52.0 | **77.5** |
| | CRISPEDIT ($\gamma = 0.6$) | 77.8 | 67.8 | 56.0 | 48.0 | **69.5** | 70.2 | **51.0** | 53.5 | **75.5** |
| | CRISPEDIT ($\gamma = 0.7$) | **80.5** | 69.0 | 57.4 | **50.9** | **69.5** | 67.9 | **50.5** | 55.0 | **76.0** |
| | CRISPEDIT ($\gamma = 0.8$) | **80.3** | 68.3 | 52.3 | 46.0 | **69.2** | 66.7 | **50.1** | 56.0 | **77.0** |
| | CRISPEDIT ($\gamma = 0.9$) | 71.2 | 57.9 | 48.0 | 40.3 | **69.4** | 68.4 | **50.3** | **59.5** | 73.0 |
| | CRISPEDIT ($\gamma = 0.95$) | 62.7 | 48.5 | 38.5 | 31.7 | **69.4** | 68.4 | **50.4** | 56.0 | **76.0** |
| | CRISPEDIT ($\gamma = 0.99$) | 37.8 | 28.8 | 35.3 | 27.6 | **69.4** | 68.9 | **51.1** | **57.5** | 73.0 |
| CounterFact | LLaMA-3-8B-Instruct | 1.2 | 1.0 | 0.3 | 0.6 | 69.5 | 69.3 | 50.7 | 58.0 | 73.5 |
| | CRISPEDIT ($\gamma = 0.5$) | 45.5 | 31.5 | 5.7 | 7.2 | **68.5** | 50.4 | **51.4** | 50.0 | 43.5 |
| | CRISPEDIT ($\gamma = 0.6$) | 65.5 | 48.7 | 9.8 | 14.3 | **69.7** | 63.2 | **52.4** | **55.5** | **75.0** |
| | CRISPEDIT ($\gamma = 0.7$) | 75.7 | 57.3 | 15.5 | 20.4 | **69.4** | 66.8 | **51.7** | 55.0 | 72.5 |
| | CRISPEDIT ($\gamma = 0.8$) | **79.2** | 57.4 | 21.4 | 25.8 | **69.5** | 69.4 | 49.8 | 54.5 | 73.5 |
| | CRISPEDIT ($\gamma = 0.9$) | **79.4** | 55.9 | 38.4 | **32.4** | **69.3** | 67.5 | 49.5 | 54.0 | **76.5** |
| | CRISPEDIT ($\gamma = 0.95$) | 72.0 | 47.5 | **46.3** | **33.0** | **69.4** | 67.8 | **50.3** | **57.0** | **74.0** |
| | CRISPEDIT ($\gamma = 0.99$) | 51.6 | 27.7 | **45.3** | 26.8 | **69.4** | 68.2 | **51.7** | 56.0 | **76.5** |
| WikiBigEdit | LLaMA-3-8B-Instruct | 9.3 | 9.1 | 16.4 | 16.1 | 69.5 | 69.3 | 50.7 | 58.0 | 73.5 |
| | CRISPEDIT ($\gamma = 0.5$) | 62.6 | 58.7 | 14.3 | 14.9 | **69.0** | 68.8 | **50.6** | 55.0 | 72.5 |
| | CRISPEDIT ($\gamma = 0.6$) | 66.5 | 60.8 | 17.4 | 19.1 | **69.3** | 68.2 | **51.4** | 53.0 | **75.0** |
| | CRISPEDIT ($\gamma = 0.7$) | **76.2** | 69.2 | 26.3 | 27.8 | **69.2** | 68.8 | **51.1** | 54.0 | **76.5** |
| | CRISPEDIT ($\gamma = 0.8$) | **77.2** | **72.1** | 21.2 | 24.4 | **69.4** | 69.1 | **50.4** | 55.0 | **76.5** |
| | CRISPEDIT ($\gamma = 0.9$) | **77.0** | 70.2 | 28.4 | 30.5 | **69.3** | 70.5 | **51.8** | 55.0 | **74.0** |
| | CRISPEDIT ($\gamma = 0.95$) | **76.9** | 68.9 | 23.4 | 27.3 | **69.2** | 62.6 | **51.2** | **57.5** | **74.5** |
| | CRISPEDIT ($\gamma = 0.99$) | 67.6 | 57.2 | **34.4** | 32.3 | **69.3** | 62.5 | **52.6** | **58.0** | 70.5 |

