# OpenReview forum: "CrispEdit: Low-Curvature Projections for Scalable Non-Destructive LLM Editing"
_ICML.cc/2026/Conference — ICML 2026 regular_

### Official Review · Reviewer_n68T · 2026-03-06

**Soundness:** 3
**Presentation:** 4
**Significance:** 3
**Originality:** 3
**Overall Recommendation:** 5
**Confidence:** 3

**Summary:**

This paper introduces CRISPEDIT, a model editing algorithm that treats capability preservation as an explicit hard constraint by projecting edit updates onto the low-curvature subspace of the capability loss landscape. To avoid assuming base model convergence, the capability constraint is instantiated via Bregman divergence, which yields a quadratic form governed by the Gauss-Newton Hessian without requiring the gradient to vanish at initialization. The authors prove that existing methods such as AlphaEdit operate within a strictly more restrictive subspace, as their activation-based null-space is a subset of the GNH null-space. To scale to billion-parameter models, CRISPEDIT approximates the GNH via K-FAC and introduces a matrix-free projector exploiting Kronecker structure. A sequential variant maintains online K-FAC statistics to protect both base capabilities and previously edited knowledge without growing memory. Experiments on LLaMA-3-8B under the WILD autoregressive evaluation protocol demonstrate strong edit performance while keeping capability degradation below 1% on average, substantially outperforming prior methods.

**Compliance With Llm Reviewing Policy:**

Affirmed.

**Final Justification:**

After reading the rebuttal, I believe the authors have adequately addressed my main concerns. Overall, I am satisfied that the paper is technically sound and that the previously raised issues have been resolved.

**Key Questions For Authors:**

See Weaknesses.

**Limitations:**

Yes

**Strengths And Weaknesses:**

**Strengths**

1. The paper is clearly written and well-organized. The progression from problem formulation to theoretical analysis to scalable implementation follows a coherent structure, and the connection between Bregman divergence, GNH, and low-curvature projections is explained with sufficient clarity.

2. The work is well-motivated with strong theoretical grounding. Proposition 1 provides a formal justification for why activation-based null-space methods such as AlphaEdit are suboptimal, showing that their feasible subspace is strictly contained within that of CRISPEDIT.

**Weaknesses**

1. The choice of energy threshold $\gamma$ has a non-trivial impact on performance but requires dataset-specific tuning.

2. The gap between QA Context and No Context reliability is substantial across all datasets, for example 80.5% vs 57.4% on ZsRE. This suggests that edited knowledge is not reliably internalized and remains partially dependent on contextual scaffolding?

3. The paper claims that K-FAC statistics can be precomputed once and reused across all edits. While the small-scale experiments in Section 4.1 explicitly recompute the projector when parameter changes exceed 25%, no analogous discussion is provided for the LLM-scale experiments.

---

> ### Author Rebuttal · Authors · 2026-03-31
>
> Thank you, Reviewer n68T, for the positive assessment and thoughtful feedback. We appreciate your recognition that the paper is “well-motivated with strong theoretical grounding” and that it provides “a formal justification for why activation-based null-space methods such as AlphaEdit are suboptimal.” In response to your comments, we ran additional experiments on **tuning and robustness of energy threshold $\gamma$**, verifying that the energy threshold $\gamma$ selected on a held-out validation set remains stable across multiple random edit data subsets. Below, we respond to your comments and questions in more detail.
>
> > "The choice of energy threshold $\gamma$ has a non-trivial impact on performance but requires dataset-specific tuning."
>
> We clarify that $\gamma$ can be tuned using a held-out validation set from each dataset. In particular, we select a **held-out, disjoint** subset of the edit data as a validation set, edit the model with various parameters, and obtain the best-performing $\gamma$. We use the same gamma for different edit batch sizes (e.g., 3K vs. 10K).
>
> Below, we run additional experiments to verify that our hyperparameter choices on validation perform well on multiple, disjoint test sets. Specifically, we fix our validation hyperparameter choice $\gamma = 0.7$, and edit on 4 disjoint, randomly selected subsets of size 10K from ZsRE. The following table demonstrates that our results with fixed hyperparameters are consistent and robust:
>
> | Subset | QA (Rel) | QA (Gen) | MMLU | IFEval | TruthfulQA | ARC-C | GSM8k |
> | :--- | :--- | :--- | :--- | :--- | :--- | :--- | :--- |
> | 1 | 0.78 | 0.72 | 67.9 | 76.0 | 50.2 | 53.0 | 79.0 |
> | 2 | 0.77 | 0.71 | 67.9 | 72.4 | 50.8 | 54.0 | 78.0 |
> | 3 | 0.75 | 0.68 | 68.0 | 76.7 | 53.1 | 51.0 | 83.0 |
> | 4 | 0.73 | 0.68 | 68.6 | 77.9 | 47.5 | 54.0 | 74.0 |
>
> > "The gap between QA Context and No Context reliability is substantial across all datasets, for example 80.5% vs 57.4% on ZsRE. This suggests that edited knowledge is not reliably internalized and remains partially dependent on contextual scaffolding?"
>
> We agree that there is a gap between QA and No Context but believe it does not indicate that the edited knowledge is not internalized. First, we clarify that the WILD evaluation protocol (Yang et al., 2025) only includes contextual evaluation with the QA Context as that provides question-answering instruction to the model. In this paper, we include the No Context result for the thoroughness of our evaluation. Note that the No Context is a harder elicitation setting: the model must recall the fact and enter the question-answering mode (as without a template or context, the instruction-finetuned model may revert back to a base, simple text-completion behavior). In contrast, the QA Context provides a question-answer framing. Importantly, the edited knowledge is still present in the No Context setting. For example, on ZsRE, No Context reliability increases from 2.9% for the unedited Llama-3-8B-Instruct model to 57.4% after applying CrispEdit. We therefore interpret the gap primarily as an elicitation gap, not a failure of internalizing the edit. Furthermore, this behavior is also consistent with our capability-preservation objective: because CrispEdit explicitly preserves the base model’s capabilities and broader behavior, it is less likely to change model behavior in under-specified prompts and when the model tends to revert to base behavior, while still achieving strong gains in both context and no-context settings.
>
> > "The paper claims that K-FAC statistics can be precomputed once and reused across all edits. While the small-scale experiments in Section 4.1 explicitly recompute the projector when parameter changes exceed 25%, no analogous discussion is provided for the LLM-scale experiments."
>
> You are correct that, in the MNIST experiments, we recompute the projector once parameter changes exceed 25%. However, we clarify that K-FAC statistics are **not** recomputed for any LLM-scale experiments. The reason is that, in small models, the smaller parameter space is more sensitive to weight updates. Thus, the curvature estimate can become stale relatively quickly after editing, since the model moves farther from the local region where the Fisher information and its K-FAC approximation become accurate. Thus, we adopted a trust-region style update. In contrast, for the LLM experiments, we found that the precomputed K-FAC statistics remained a sufficiently stable approximation throughout editing, so recomputation was unnecessary in practice (empirical evidence:  https://imgur.com/a/PU6TJ88). This is consistent with the larger overparameterization of LLMs, where local curvature tends to vary less sharply under the relatively small parameter updates done through editing. We will clarify this distinction in the revised manuscript.
>
>  References
>
> Yang, Wanli, et al. "The mirage of model editing: Revisiting evaluation in the wild." ACL 2025.

---

> > ### Author Rebuttal · Reviewer_n68T · 2026-04-02
> >
> > Thanks for your detailed rebuttal.

---

> > > ### Author Response · Authors · 2026-04-04
> > >
> > > Thank you for your acknowledgement. We are very glad to know that our rebuttal has adequately addressed your concerns and that the issues you raised are now fully resolved. We are truly grateful for your support of our work.

---

### Official Review · Reviewer_3kj4 · 2026-03-13

**Soundness:** 3
**Presentation:** 3
**Significance:** 3
**Originality:** 3
**Overall Recommendation:** 4
**Confidence:** 3

**Summary:**

This paper studies model editing. It proposes CRISPEDIT, a constrained editing method that projects parameter updates into low-curvature directions of the capability loss, so the model can absorb targeted edits while better preserving its original behaviors. The method is further made practical for LLMs through a Bregman–Newton formulation, K-FAC approximation, and a matrix-free projector. Empirically, the paper shows a strong edit–capability trade-off on standard editing benchmarks, and the method also extends to sequential and larger-scale editing settings.

**Compliance With Llm Reviewing Policy:**

Affirmed.

**Key Questions For Authors:**

Please refer to weakness.

**Limitations:**

yes

**Strengths And Weaknesses:**

### Strength
1. I like that the paper also clarifies the relation to prior work. It also explains why methods like AlphaEdit can be seen as more restrictive special cases.

2. The paper includes sequential editing, scaling to more edits, and a cross-model test on Qwen. These make this paper compatible.

### Weakness
1. The paper lacks enough details to replicate the paper. For example, the paper does not clearly describe the actual estimation procedure of the K-FAC statistics in transformers, including token selection, averaging scheme, numerical stabilization.

2. The paper edit only a fixed set of MLP down-projection layers, but the paper does not explain how these layers were selected.

---

> ### Author Rebuttal · Authors · 2026-03-31
>
> Thank you, Reviewer 3kj4, for the positive assessment of our work, including noting that “the paper clarifies the relation to prior work” and that it “includes sequential editing, scaling to more edits, and a cross-model test on Qwen.”
>
> Per your suggestions and in response to your comments, we have **expanded our methodological details and ran new experiments**: (1) *enhanced reproducibility:* providing our **complete anonymized codebase** and detailing the *exact estimation procedure* for the K-FAC statistics, including token selection and numerical stabilization; (2) *layer selection ablation:* conducting **additional experiments across multiple different layer subsets** on **Qwen-2.5 1.5B and 32B** models to empirically justify our layer choices. Below, we respond to your comments and questions in more detail.
>
> > “The paper lacks enough details to replicate the paper. For example, the paper does not clearly describe the actual estimation procedure of the K-FAC statistics in transformers, including token selection, averaging scheme, numerical stabilization.”
>
> Thank you for your feedback, and we agree that reproducibility is important. We clarify that our submission already includes the full algorithmic description of CRISPEdit and CRISPEdit-SEQ, K-FAC cache construction, evaluation protocols, and LLM judge prompt, and hyperparameters and model-specific settings in Appendix G.
>
> To aid complete reproducibility, **we are providing our complete anonymized** codebase to reproduce our results in the following link: [https://anonymous.4open.science/r/CrispEdit](https://anonymous.4open.science/r/CrispEdit)
>
> *Explanation of the K-FAC estimation procedure* We provide more discussion on our approach for estimating K-FAC. The K-FAC statistics are computed over Wikipedia text using the standard next-token prediction objective. For each sequence in the corpus, forward and backward hooks are registered on the target MLP `down_proj` layer to capture (i) the layer's input activations and (ii) the gradient of the cross-entropy loss with respect to the layer's output. Only non-padding tokens contribute: a validity mask (`labels != -100`) is applied to both the input activation matrix and the output gradient matrix before accumulating the outer-product sums $A$ (input covariance) and $B$ (gradient covariance), which are then normalized by the total number of valid tokens. Numerical stability in the subsequent SVD is ensured by using `torch.linalg.eigh` on the symmetric positive semi-definite factors in float64 precision; the null-space rank is determined by an energy threshold on the sorted eigenvalue spectrum (i.e., the smallest eigenvectors whose cumulative energy falls below `1 − τ` are retained as the null-space basis). We will add further details on our K-FAC estimation procedure to the paper.
>
> > “The paper edit only a fixed set of MLP down-projection layers, but the paper does not explain how these layers were selected.”
>
> Thank you for pointing this out. We clarify our layer-selection rationale below and will add it to the paper. Our choice to edit a fixed set of **MLP down-projection layers** follows the standard practice of editing methods, which target MLP blocks as they have been shown to act as key-value memories and are important in factual recall. For instance, ROME (Meng et al., 2022) identifies middle-layer FFNs important for factual prediction and MEMIT (Meng et al., 2023) edits a critical range of MLP layers selected via causal tracing. Likewise, AlphaEdit uses a fixed set of contiguous layers following MEMIT.
>
> In our implementation, we therefore adopted a small fixed contiguous window of MLP layers for each model, for computational efficiency and fair comparison to prior work. To pick the window of layers, we ran sweeps on a validation set and picked the best-performing window, along with following the layer choice recommended by Loc-BF-FT (Yang et al., 2026), a baseline that only fine-tunes one layer for model editing by treating it as a hyperparameter, ensuring that the layer is included.
>
> We present **new experimental results** on Qwen-2.5 1.5B and Qwen-2.5 32B: https://imgur.com/a/Br6slg8 Our empirical results suggest that layers at either 25% or 75% of model depth (with additional layers around them) tend to perform better, which is aligned with Loc-BF-FT (Yang et al., 2026) findings. We will add this rationale and the new ablation results to the revision.
>
>
> ###### References:
>
> ###### Yang, Wanli, et al. "Fine-tuning Done Right in Model Editing." ICLR 2026
>
> ###### Meng, Kevin, et al. "Locating and editing factual associations in gpt." NeurIPS 2022
>
> ###### Meng, Kevin, et al. "Mass-editing memory in a transformer." ICLR 2023

---

> > ### Author Rebuttal · Reviewer_3kj4 · 2026-04-03
> >
> > Thanks for a detailed rebuttal. I will keep my positive score.

---

> > > ### Author Response · Authors · 2026-04-04
> > >
> > > Thank you for your acknowledgement. We are very glad to know that our rebuttal has adequately addressed your concerns and that the issues you raised are now fully resolved. In light of your updated assessment, we would be very grateful if you would consider revising your initial score accordingly.

---

### Official Review · Reviewer_6KwY · 2026-03-15

**Soundness:** 3
**Presentation:** 3
**Significance:** 2
**Originality:** 3
**Overall Recommendation:** 3
**Confidence:** 3

**Summary:**

This study focuses on an interesting challenge. Addressing the problem of preserving capabilities during the editing process of large language models, which is a core issue: traditional methods often sacrifice the model's general capabilities while achieving successful edits. The authors raise a key question: how can capability preservation be enforced directly without converting the editing into a full retraining?
This research proposes the CrispEdit framework, an in-place parameter editing framework with curvature constraints, which models the editing problem as a constrained optimization problem. The core ideas include projecting updates onto a low-curvature subspace of the capability loss landscape, using Bregman divergence to avoid assumptions about the convergence of the base model, and leveraging Kronecker factorized approximations of curvature (K-FAC) to achieve efficiency at the scale of large language models.
Testing Models are  LLaMA-3-8B-Instruct and  Qwen-2.5-1.5B-Instruct. Datasets of 3,000 edits on each of three standard datasets. Experiments show good performance improvements.

**Compliance With Llm Reviewing Policy:**

Affirmed.

**Key Questions For Authors:**

1.  The paper proposes the use of Bregman divergence to avoid assumptions about the convergence of foundational models. Could you elaborate more on how this method adapts to different convergence states during the actual training process of LLMs?
2.  Regarding the impact of the regularization parameter  on the trade-off between editing success rate and capability retention, could you provide a more detailed discussion on the balance between editing success rate and capability retention for different parameter choices?
3.  The paper mentions that performing 3000 edits takes 6 hours on an A40 GPU. Could you explain the storage requirements for the model parameters after these edits, and how much this increases compared to the storage of an unedited model?

**Limitations:**

yes

**Strengths And Weaknesses:**

- **Strengths**:
  - The theoretical derivation is comprehensive. By reformulating the optimization framework and using Bregman divergence to derive low-curvature projections, the method's rationality is thoroughly demonstrated, providing a more thorough theoretical analysis compared to methods like AlphaEdit.
  - The experimental design is comprehensive, validating the theory on small-scale datasets and extending to large-scale LLM editing tasks. The evaluation metrics include edit success rate, generalization ability, and the retention of general performance (e.g., MMLU, GSM8K, IFEval), ensuring a systematic and complete assessment.
  - A thorough ablation study is conducted, comparing different variants such as LocBF-FT, AlphaEdit, and MEMIT, accurately identifying the advantages of CrispEdit in the trade-off between editing and capability.
- **Weaknesses**:
  - The discussion on the robustness of hyperparameter choices, such as the curvature threshold \gamma, is adequate, but a deeper analysis on how to select \(\gamma\) for different problem sizes would benefit the readers.
  - Sequential editing of 3000 edits on a single A40 GPU still takes about 44 minutes, which is slower than batch editing; while batch processing of 3000 edits can be completed in just 3-6 minutes, which is highly efficient compared to alternatives.
- Only 8B and 1.5 B models are experimented, if more different sizes models are tested, experiments are more soundful.

---

> ### Author Rebuttal · Authors · 2026-03-31
>
> Thank you, Reviewer 6KwY, for your thoughtful feedback and for recognizing the strengths of our work, noting that “theoretical derivation is comprehensive”, “method's rationality is thoroughly demonstrated”, and that “experimental design is comprehensive”. In response, we ran **new experiments**: (1) across **different convergence stages**; (2) on a **large 32B model**; and (3) demonstrating **hyperparameter robustness across data subsets**.
>
> >"a deeper analysis on how to select (\gamma) for different problem sizes would benefit the readers."
>
> We select hyperparameters, including $\gamma$, by finding the best values on a held-out, disjoint validation set. We then fix these hyperparameters to edit the model on the test set for different edit batch sizes (e.g., 3K vs. 10K). To verify robustness, we fix the validation-selected value $\gamma = 0.7$ and evaluate performance on four disjoint 10K subsets of ZsRE. Results are consistent across subsets, suggesting that the selected $\gamma$ generalizes well: https://imgur.com/a/Z0k5CR9
>
> > "Sequential editing of 3000 edits on a single A40 GPU still takes about 44 minutes, which is slower than batch editing"
>
> Sequential editing is inherently slower than batch editing because edits must be incorporated online and chunk-wise, which prevents leveraging parallel processing. We consider this setting as it reflects realistic scenarios and allows us to study forgetting over successive edits. In our setup, 3k edits are processed as 30 chunks of 100 (∼\sim∼1.3 minutes/chunk). Importantly, *CrispEdit-Seq is the fastest sequential method we evaluate*, and is *10x faster than AlphaEdit*.
>
> > "Only 8B and 1.5 B models are experimented, if more different sizes models are tested, experiments are more soundful."
>
> We ran new experiments on Qwen-2.5-32B-Instruct on ZsRE, CounterFact, and WikiBigEdit. Full table: https://imgur.com/a/tENXdpg
>
> Results show that CrispEdit remains effective at edit success in large models while preserving base capabilities, whereas representation-constrained baselines such as AlphaEdit and Adam-NSCL perform poorly (often <10% edit success) despite extensive hyperparameter tuning.
>
> > “The paper proposes the use of Bregman divergence to avoid assumptions about the convergence of foundational models. Could you elaborate more on how this method adapts to different convergence states during the actual training process of LLMs?”
>
> Theoretically, using the standard Euclidean distance in the constraint gives a second-order expansion with a linear term $\langle \nabla L_{\text{cap}}(\theta_0), \Delta \theta \rangle$, so a low-curvature projection is justified only near convergence. Bregman divergence removes this requirement as it is *exactly first-order flat*, regardless of the convergence state/checkpoint, and reduces to a quadratic form in the Gauss-Newton Hessian without stationarity assumption. Thus, CrispEdit automatically adapts to the model convergence states as the same projection principle applies at early, intermediate, and late checkpoints. We also verified this empirically:
>
> **1. Small-scale:** We pre-train LeNet-5 on MNIST to three convergence levels, then fine-tune each on Fashion-MNIST. CrispEdit preserves most pre-training accuracy even from early, minimally-converged checkpoints: https://imgur.com/a/ui4SQDU
>
> **2. LLM:** We evaluate CrispEdit on OLMo-1B checkpoints spanning multiple convergence stages. Since OLMo-1B is not instruction-tuned, we measure base capability on ARC-Challenge, MMLU, and HellaSwag, and assess edit quality with teacher-forced rewrite/rephrase metrics: https://imgur.com/a/dm2Gf7W
>
> Across all checkpoints, rewrite accuracy remains >98%. While rephrase accuracy is lower at the earliest checkpoint (63% vs. ~93% for later), this mainly suggests that very early-stage models generalize less. Importantly, the base capability declines only slightly after editing, indicating that CrispEdit preserves capability robustly across convergence states.
>
> > "could you provide a more detailed discussion on the balance between editing success rate and capability retention for different parameter choices?"
>
> Table 8 (Appx. I) in our submission shows the editability-preservation trade-off controlled by $\gamma$: smaller $\gamma$ allows larger updates and improves edit success, while larger $\gamma$ better preserves the base model but can make edits too conservative.  For example, on ZsRE, Reliability drops from 80.5 at $\gamma=0.7$ to 37.8 at $\gamma = 0.99$. In practice, $\gamma \in [0.7, 0.9]$ provides a good balance. Additional plots to illustrate edit-capability trade-off: https://imgur.com/a/LdCxhE6
>
> > "The paper mentions that performing 3000 edits takes 6 hours on an A40 GPU. Could you explain the storage requirements for the model parameters after these edits…?"
>
> We clarify that performing 3000 edits takes 6 minutes on an A40 GPU. Furthermore, *CrispEdit also incurs no additional storage cost*, since it directly fine-tunes the model parameters.

---

### Official Review · Reviewer_HbND · 2026-03-18

**Soundness:** 3
**Presentation:** 4
**Significance:** 3
**Originality:** 4
**Overall Recommendation:** 4
**Confidence:** 4

**Summary:**

The authors present CrispEdit, a scalable and principled second-order editing algorithm that treats capability preservation as an explicit constraint, unifying and generalizing several existing editing approaches. CrispEdit formulates editing as constrained optimization and enforces the constraint by projecting edit updates onto the low-curvature subspace of the capability-loss landscape. The authors propose to make second-order procedure efficient at the LLM scale using Kronecker-factored approximate curvature (K-FAC) and a novel matrix-free projector that exploits Kronecker structure to avoid constructing massive projection matrices.

**Compliance With Llm Reviewing Policy:**

Affirmed.

**Key Questions For Authors:**

1. The authors are encouraged to experiment their proposed approach on other open-source models like Qwen-2.5 series or Phi model series to show their claims are generalizable to different LLMs.
2. The authors are encouraged to provide qualitative examples of success and failure cases they observe of model capability / collapse in baselines compared to their method.
3. Reproducibility study is limited that will make replicating this study tough.

**Limitations:**

1. The authors are encouraged to experiment their proposed approach on other open-source models like Qwen-2.5 series or Phi model series to show their claims are generalizable to different LLMs.
2. The authors are encouraged to provide qualitative examples of success and failure cases they observe of model capability / collapse in baselines compared to their method.
3. Reproducibility study is limited that will make replicating this study tough.

**Strengths And Weaknesses:**

1. The paper is well written and organized.
2. The authors solve a relevant problem of LLM behavior editing without losing existing capabilities.
3. I appreciate the core argument of the paper on the lines of model editing as a constrained optimization problem to minimize the edit loss subject to negligible changes in capability loss.
4. The authors perform exhaustive experiments to support their claims.

---

> ### Author Rebuttal · Authors · 2026-03-31
>
> Thank you, Reviewer HbND, for the positive assessment of our work, including your comment that you “appreciate the core argument of the paper” and that the “authors perform exhaustive experiments to support their claims”. While we have already provided experimental results on two models, Llama-3 8B and Qwen-2.5 1.5B, based on your suggestion, we ran **new experiments** on **Qwen 2.5 32B on three datasets** and **Qwen-2.5 1.5B on two additional datasets**. Furthermore, we provide an **anonymous link to our code** to support reproducibility and present qualitative examples. We respond to your comments and provide more details below.
>
> > The authors are encouraged to experiment their proposed approach on other open-source models like Qwen-2.5 series or Phi model series to show their claims are generalizable to different LLMs.
>
>
> Thank you for your suggestion. We would like to clarify that we had already included experiments on two model families: Llama-3 8B Instruct (main paper) and Qwen-2.5 1.5B Instruct (Appendix I, Table 5), showing that our results are generalizable and robust across model families.
>
> Following your suggestion, we ran **additional experiments,** demonstrating the robustness of our approach across model sizes and families:
> 1. **Qwen-2.5 1.5B Instruct** on CounterFact and WikiBigEdit, in addition to the ZsRE results already included in the paper.
> 2. **Qwen-2.5 32B Instruct** on all three datasets ZsRE, CounterFact, and WikiBigEdit.
>
> Due to the character limit, we only provide ZsRE results for the 32B model below. Our **complete tables are presented here**: https://imgur.com/a/lNm297K and https://imgur.com/a/vUn5x0L
>
> Our new experiments further strengthen the claims of our work, showing that CrispEdit remains effective across model families and scales to substantially larger models, while preserving base capabilities. In particular, on Qwen-2.5 32B Instruct, CrispEdit continues to achieve strong performance, whereas representation-constrained methods such as AlphaEdit and Adam-NSCL show poor performance (e.g., often below 10% edit success across settings) despite extensive hyperparameter search. We will add the new results to the revised manuscript.
>
> **Qwen-2.5 32B Instruct**
>
> | Data | Model | QA Context (Rel) | QA Context (Gen) | No Context (Rel) | No Context (Gen) | MMLU | IFEval | TruthfulQA | ARC-C | GSM8K |
> | :--- | :--- | :--- | :--- | :--- | :--- | :--- | :--- | :--- | :--- | :--- |
> | **ZsRE** | Qwen2.5-32B Base | 3.1 | 3.1 | 2.7 | 2.3 | 84.3 | 80.0 | 65.0 | 71.5 | 82.0 |
> | | CrispEdit (L13-17) | **78.3** | **63.3** | 21.6 | 18.3 | **83.8** | **79.3** | **65.1** | **71.5** | **82.0** |
> | | Adam-NSCL | 9.1 | 7.7 | 11.5 | 5.9 | **84.1** | 67.2 | **62.9** | **74.0** | **81.0** |
> | | AlphaEdit | 0.0 | 0.0 | 0.0 | 0.0 | 22.9 | 18.2 | 51.5 | 25.5 | 0.0 |
> | | LocBF-FT | 61.7 | 43.9 | **30.5** | **21.2** | **83.7** | **77.5** | **63.7** | 69.5 | **80.5** |
> | | UltraEdit | 17.4 | 14.5 | 22.4 | 16.2 | **84.2** | **79.2** | **63.6** | 68.0 | **80.5** |
>
> > “The authors are encouraged to provide qualitative examples of success and failure cases they observe of model capability / collapse in baselines compared to their method.”
>
> We have presented some qualitative examples in Appendix H of our submission, showing that other editors can result in model collapse, as evidenced by the repetition in response. As indicated by our main result in **Table 1**, baselines such as AlphaEdit, MEMIT, AdamNSCL, and FT can suffer from capability degradation and collapse, while our approach, CrispEdit, succeeds.
>
> In addition to model collapse, we qualitatively also observe degradation of capability compared to the base model. For instance, the model edited by AlphaEdit fails to properly reason in answering math questions, whereas the model edited by our approach, CrispEdit, successfully performs reasoning. We provide a Table of such qualitative examples on GSM8K in this link: https://imgur.com/a/RhJp1yb
>
> > “Reproducibility study is limited that will make replicating this study tough.”
>
> Thank you for your feedback; we agree that reproducibility is important. We would like to clarify that our submission already includes the full algorithmic description of CRISPEdit and CRISPEdit-SEQ, KFAC cache construction, evaluation protocols and LLM judge prompt, and hyperparameters and model-specific settings in Appendix G.
>
> To further support reproducibility, **we are also providing a complete anonymized codebase** that reproduces our experiments: https://anonymous.4open.science/r/CrispEdit

---

> > ### Author Rebuttal · Reviewer_HbND · 2026-04-03
> >
> > Thanks for a detailed rebuttal.

---

> > > ### Author Response · Authors · 2026-04-04
> > >
> > > Thank you for your acknowledgement. We are very glad to know that our rebuttal has adequately addressed your concerns and that the issues you raised are now fully resolved. In light of your updated assessment, we would be very grateful if you would consider revising your initial score accordingly.

---

### Decision · Program_Chairs · 2026-04-30

**Decision:**

Accept (regular)

**Comment:**

Reviewer 6KwY did not provide any responses to the authors rebuttal so I omitted the review. Thus, all three reviewers agree to accept the paper. I have read the paper and found that it indeed provide a new perspective than previous work on KE. Thus, it should be accepted.